# Retrieval of snow and soil properties for forward radiative transfer modeling of airborne Ku-Band SAR to estimate snow water equivalent: The Trail Valley Creek 2018/19 Snow Experiment

Benoit Montpetit[1], Joshua King[1,†], Julien Meloche[1], Chris Derksen[1], Paul Siqueira[2], J. Max Adam[2], Peter Toose[1], Mike Brady[1], Anna Wendleder[3], Vincent Vionnet[4], and Nicolas R. Leroux[4]

[1]Climate Research Division, Environment and Climate Change Canada, Ontario, Canada
[2]College of Engineering, University of Massachusetts Amherst, MA, United States
[3]German Aerospace Center (DLR), German Remote Sensing Data Center, Oberpfaffenhofen, Germany
[4]Meteorological Research Division, Environment and Climate Change Canada, Quebec, Canada
[†]deceased, 21 February, 2023

**Correspondence:** Benoit Montpetit (benoit.montpetit@ec.gc.ca)

**Abstract.** Accurate snow information at high spatial and temporal resolution is needed to support climate services, water resource management, and environmental prediction services. However, snow remains the only element of the water cycle without a dedicated Earth Observation mission. The snow scientific community has shown that Ku-Band radar measurements provide accurate snow information with its sensitivity to snow water equivalent and the wet/dry state of snow. Recently de-
veloped tools like the Snow MicroPenetrometer (SMP), to retrieve snow microstructure data in the field, and radiative transfer models like the Snow Microwave Radiative Transfer Model (SMRT), are promising tools for characterizing snow and how it translate into radar backscatter. An experiment at Trail Valley Creek (TVC), Northwest Territories, Canada was conducted during the winter of 2018/19 in order to characterize the impacts of varying snow geophysical properties on Ku-Band radar backscatter at a 100-m scale. Airborne Ku-Band data was acquired using the University of Massachusetts radar instrument.
This study shows that it is possible to calibrate SMP data to retrieve statistical information on snow geophysical properties and properly characterize a representative snowpack at the experiment scale. The tundra snowpack measured during the campaign can be characterized by two layers corresponding to a rounded snow grain layer and a depth hoar layer. Using Radarsat-2 and TerraSAR-X data, soil background roughness properties were retrieved ($mss_{\text{soil}} = 0.010 \pm 0.002$) and it was shown that a single value could be used for the entire domain. Recently, it was shown that snow grain size, represented by exponential correlation
length, could be translated to its equivalent *snow microwave grain size* by a parameter called polydispersity. Values of 0.74 and 1.11 for rounded and depth hoar snow grains polydispersity, respectively, was retrieved. Using the Geometrical Optics surface backscatter model, the retrieved effective soil permittivity increased from C-Band ($\varepsilon_{\text{soil}} = 2.47$) to X-Band ($\varepsilon_{\text{soil}} = 2.61$), to Ku-Band ($\varepsilon_{\text{soil}} = 2.77$) for the TVC domain. Using SMRT and the retrieved soil and snow parameterizations, an RMSE of 2.6 dB was obtained between the measured and simulated Ku-Band backscatter values when using a global set of parameters for
all measured sites. When using a distributed set of soil and snow parameters, the RMSE drops to 0.9 dB. This study thus shows that it is possible to link Ku-Band radar backscatter measurements to snow conditions on the ground using a priori knowledge of the snow conditions, which is of great importance in SWE retrieval algorithms from Ku-Band SAR measurements.

# 1 Introduction

Snow is an important freshwater resource and remains the only element in the water cycle without a dedicated spaceborne mission (Derksen et al., 2019). While surface snow depth observation networks support the generation and validation of coarse resolution (>25 km) snow water equivalent (SWE) products from passive microwave remote sensing (e.g., Luolus et al., 2021), higher spatial resolution (<500m) SWE products are needed to meet the needs of climate services, water resource management, and environmental prediction (Garnaud et al., 2019, 2021; Kim et al., 2021; Cho et al., 2023).

Tower and airborne measurements (Lemmetyinen et al., 2018; Zhu et al., 2021) supported by theoretical modeling (e.g., Xu et al., 2012; Tsang et al., 2007) show that Ku-band radar measurements (13.5 and 17.25 GHz) provide a viable pathway for a future satellite mission capable of monitoring snow water storage because of (1) sensitivity to SWE through the volume scattering properties of dry snow and (2) the ability to discriminate wet from dry snow cover (Tsang et al., 2022), as a complement to existing C-Band SAR methods (Stiles and Ulaby, 1980).. While Ku-band radar measurements are available from altimetry and precipitation missions (CryoSAT-2 and Sentinel-3, CloudSat) there are no current SAR missions at this frequency available for science applications. Despite limited availability of measurements, significant progress has been made over the past decade in understanding of Ku-band radar response to SWE, snow microstructure, and snow wet/dry state in support of past and current mission concepts (e.g., Tsang et al., 2022; Derksen et al., 2019; Rott et al., 2010).

Advancement of a Ku-band radar-based SWE retrieval is highly dependent on decoupling the strong spatial and seasonal influences of snow microstructure and background backscatter (Picard et al., 2022; Meloche et al., 2021). Unconstrained, known variations in these properties can modify Ku-band backscatter in ways comparable to SWE in terrestrial environments, making retrievals impossible. In this paper, extensive measurements of snow microstructure and soil properties collected in a tundra environment are used to constrain known uncertainties and evaluate the capability of a forward electromagnetic model (Snow Microwave Radiative Transfer Model (SMRT), Picard et al., 2018) to reproduce observed backscatter from a new set of Ku-band (13.5 GHz) airborne measurements. We present a multi-frequency approach, in which we decouple the background soil contribution using C-band satellite data, from the snow volume scattering contribution at Ku-band. By illustrating the successful forward simulation of measured Ku-band backscatter using an open source electromagnetic model, we successfully demonstrate a crucial component of the cost function SWE retrieval concepts described in the literature (Rott et al., 2012).

While Ku-band radar measurements have clear potential for measuring SWE, experimental airborne and tower measurements are limited. Tower-based measurements at 10.2, 13.3, and 16.7 GHz were collected over four winter seasons in Sodankylä, Finland (2009/10 through 2012/13) complemented with multi-frequency passive microwave measurements of an overlapping footprint. The synergistic radiometer measurements were effective in providing first-guess information on effective snow grain size which was used within the SWE retrieval approach developed and evaluated by Lemmetyinen et al. (2018). These tower measurements have also been used to support other algorithm development experiments (Merkouriadi et al., 2021; Zhu et al., 2021; Durand et al., 2024; Pan et al., 2023), with the daily temporal resolution proving to be particularly insightful. However, a major limitation was the lack of spatial sampling, so efforts in the community shifted to the acquisition of airborne Ku-band radar data.

The European Space Agency (ESA) SnowSAR instrument was developed to support science development activities for the proposed Cold Regions Hydrology High-Resolution Observatory (CoReH2O) satellite mission (Rott et al., 2010). SnowSAR is a side-looking, dual-polarised (VV/VH), X/Ku band synthetic aperture radar (SAR), operable from various aircraft. Between 2010 and 2013, the instrument was deployed at several sites in Northern Finland, Austrian Alps, and northern Canada. These data, along with comprehensive snow measurements during the data acquisition periods, is freely available as described in Lemmetyinen et al. (2022). While temporally limited, these measurements provide the first spatially distributed Ku-band backscatter data, which provided a new perspective on radar signatures in snow covered environments, including the important influence of snow microstructure (King et al., 2018). Collective analysis of the SnowSAR datasets from the CoReH20 era showed the potential for Machine Learning (ML) techniques to be effectively trained to retrieve SWE across the range of snow-climate classes flown by SnowSAR (Santi et al., 2021). The SnowSAR was also flown over Grand Mesa, Colorado as part of the NASA Snow Experiment (SnowEx) in 2017 (Singh et al., 2023). These studies (e.g., Tsang et al., 2022; Lemmetyinen et al., 2022; King et al., 2018) have shown that the Ku-Band frequency range is most sensitive to SWE, and a priori knowledge of snow microstructure is necessary to accurately estimate SWE. This is why a Canadian, dual-frequency Ku-Band (13.5 and 17.25 GHz), dual-polarization (VV/VH) SAR mission, is currently in development by the Canadian Space Agency and Environment and Climate Change Canada (Terrestrial Snow Mass Mission, TSMM). This Canadian led mission aims at providing weekly coverage over the Northern Hemisphere at a nominal resolution of 500 m.

A dual-frequency (13.285 and 35.9 GHz) radar was developed at The University of Massachusetts (UMass) as a demonstrator and precursor to NASA's Surface Water and Ocean Topography (SWOT) mission. Subsequently, the lower frequency Ku-band component of the system at 13.285 GHz was repurposed in 2018 and developed into an airborne system that could be easily ported between common aircraft platforms. In this study, we utilized measurements from this UMass instrument, deployed during the 2018/19 winter over the Trail Valley Creek study area in the Northwest Territories, Canada, to advance science readiness activities of TSMM (Siqueira et al., 2021).

This paper focuses on the forward radiative transfer modeling component of a future SWE retrieval algorithm similar to what is proposed by (Rott et al., 2012; Pan et al., 2023). The data collected during the TVC experiment are used to validate SMRT as the forward model used for such SWE algorithm and will provide insight on the model parameterization needed to link snow properties to the Ku-Band signal, i.e. 1) isolating the background soil contribution to the backscattered signal and 2) tuning the snow microstructure to properly link the snow volume scattering to SWE.

## 2 The Trail Valley Creek 2018/19 Snow Radar Experiment

To evaluate Ku-band radar sensitivity to snow properties, airborne measurements with the UMass Ku-band radar instrument were acquired over the Trail Valley Creek (TVC) watershed near Inuvik, Northwest Territories, Canada, in November, January, and March during the 2018/19 winter season (Siqueira et al., 2021). Snow measurements and distributed hydrological modeling research activities have been conducted at TVC since 1995 (e.g., Shi et al., 2015; Wilcox et al., 2022), including a portion of the airborne SnowSAR snow radar campaign conducted during the 2012/13 winter season (King et al., 2018). Amongst these

90 studies, a detailed vegetation map was completed by Grünberg and Boike (2019). The SikSik sub-basin studied for the TVC experiment is predominantly composed of tussock tundra (39.1%), followed by dwarf shrub (30.6%) and lichen (26.6%) covers, and sparse riparian shrub elements (3.7%). Isolated black spruce stands located within the TVC research watershed were not evaluated in this study. The snowpack at TVC is a typical Canadian Arctic snowpack with a maximum snow depth around 50 cm and two dominant snow layers, i.e. a wind slab with mostly rounded snow grains at the surface and a less cohesive depth

hoar grain layer at the bottom (Rutter et al., 2019; Derksen et al., 2009).

## 2.1 Airborne SAR measurements

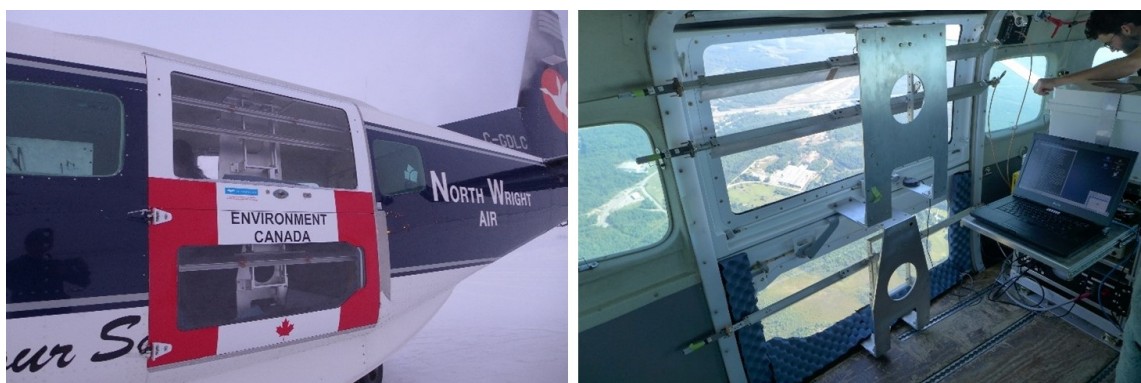

**Figure 1.** UMass Ku-band radar mounted in a modified Cessna 208 baggage door (Left). A single transmit antenna is mounted in the lower half and two receive antennas in the upper half (Right). Electronics for the system are mounted as a single rack easily transported between a variety of aircraft platforms.

For the TVC experiment, a vertically polarized waveguide antenna for signal transmission was installed in the lower half of a modified Cessna-208 baggage door (Figure 1). Two additional waveguides were mounted in the upper half of the door for simultaneous dual-channel reception, enabling both single-polarization VV SAR and single-pass InSAR capabilities. At a

100 nominal flight altitude of 1000 m, the system images a 2 km swath with a 2 m ground-range resolution and across an incidence angle range of ∼20-70°.

Flight lines with the UMass system were optimized to maximize repeat coverage of the smaller SikSik sub-basin and surrounding regions (∼24 km²) within the greater TVC watershed (58 km², Figure 2). Concentrating the flight lines over this smaller area of interest was done to maximize overlap with coincident ground snow surveys and locally installed meteoro-

105 logical and soil stations (Figure 2). Furthermore, the design of the radar acquisitions involving generous swath overlaps, and inter-campaign repeat passes were implemented to allow for filtering of motion uncertainties where needed, as well as increase inter-swath calibration opportunities (see, King et al., 2018).

Balancing budget, time, and overlap requirements, 16 flight lines were planned, all areas within the selected domain were measured in four distinct look-directions and from multiple incident angles due to the 75% overlap established between se-

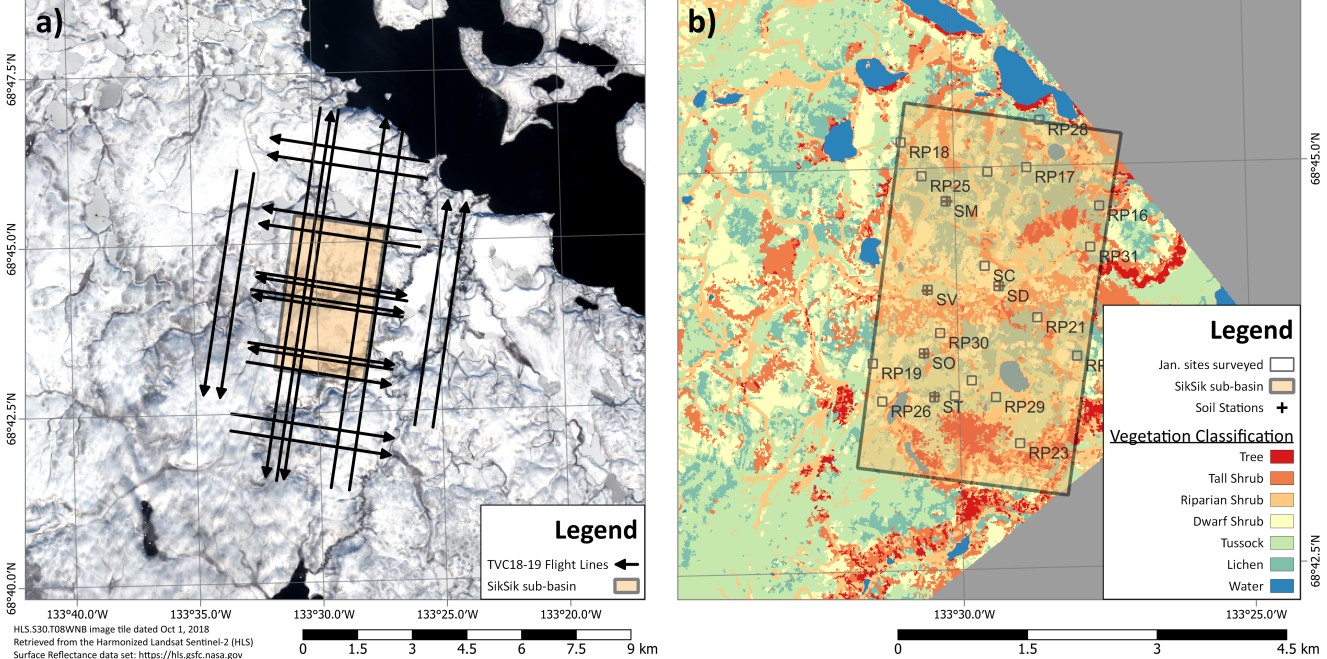

**Figure 2.** Flight lines completed during each of the TVC snow deployments (a). The 2016 vegetation classification map (Grünberg and Boike, 2019) with the location of the surveyed sites and soil stations (b). The weather station is located at the SM site. The size of the surveyed sites box corresponds to the 100 m footprint of the radar data.

quential passes. Where time allowed, flight lines were repeated within each deployment to generate further samples. Multiple revisits during a flight mission generated a diversity of radar viewing geometries.

This study will focus on optimizing the forward modelling of the SAR data acquired during the January campaign to focus on dry winter snow conditions. The November campaign Ku-Band data requires additional calibration due to unstable flight lines. This is attributed to difficult flying weather conditions and the need for manual in flight corrections from the pilot to control the roll, pitch, and yaw of the plane. The January flights had much more stable flight lines yielding better calibrated SAR data. The March campaign had above freezing temperature which made it difficult to retrieve snow properties in the field. Some liquid water content in the top portion of the snowpack was also present which reduces the influence of ground properties and snow microstructure in forward modelling of the snowpack due to the overwhelming influence of wet snow on the radar signal.

## 2.2 Satellite SAR measurements

For this campaign, C-Band satellite SAR data from RADARSAT-2 (RSAT-2) was acquired as well as X-Band TerraSAR-X (TSX) data. A total of 87 Wide Fine Quad (FQW) RSAT-2 and 40 TSX StripMap (SM) images were acquired over the

SikSik basin from September 2018 to June 2019. Table 1 gives the details of the different acquisition modes, polarizations, and incidence angles for both RSAT-2 and TSX.

**Table 1.** List of beam modes, respective polarizations, and incidence angles for the RSAT-2 and TSX acquisitions.

| Sensor | Beam Mode | Polarizations | $\theta$ range | # of scenes |
|--------|-----------|---------------|----------------|-------------|
| RSAT-2 | FQ2W | | 19.0-22.7 | 19 |
| | FQ6W | | 23.7-27.2 | 5 |
| | FQ12W | HH+HV+ | 30.6-33.7 | 11 |
| | FQ14W | VH+VV | 32.7-35.7 | 8 |
| | FQ17W | | 35.7-38.6 | 13 |
| | FQ19W | | 37.7-40.4 | 31 |
| TSX | SM | VH+VV | 33.0-34.4 | 24 |
| | | HH+HV | 38.1-39.4 | 16 |

With little sensitivity of the C- and X-Band signal intensity to snow volume scattering at co-polarizations (co-pol) (Duguay and Bernier, 2012), these satellite datasets were used to estimate the background contribution of the total backscattered signal in the forward modeling scheme (section 3.3). To focus on the UMass Ku-Band data forward modelling optimization, which is single-polarization VV, only the VV channel of the satellite imagery was used in this study. The available satellite data is provided in Table 1 as information in case future work requires additional polarizations. To increase the number of points in the optimization process, the satellite imagery acquired two weeks before and after the intensive ground campaigns were considered. Only imagery that showed intensity variability (standard deviation) below 2 dB, from one image to another, for all surveyed sites, were considered.

## 2.3 Ground based snow and soil measurements

Within the SikSik sub-basin, six static sites were established to represent the contrasting land cover and associated snow conditions also present across the greater TVC domain (Figure 2): a snow drift site (SD), a site near the meteorological station (SM), a site near an old trench site (SO Rutter et al., 2019), an open tundra site (ST) and a site within a valley (SV). Repeat snow measurements were completed during each deployment, to characterize temporal variability in snow properties. Care was taken to not complete measurements in identical locations on successive deployments, however, the general location of the sites remained the same. At these sites, four HydroProbe soil sensors were installed horizontally in a soil pit in each of the cardinal directions approximately 5 m outwards from a central datalogger. Two sensors were buried at 5 cm and two were buried 10 cm below the surface within the top layer of organic material. The soil sensor networks collected hourly measurements of temperature, moisture, and complex permittivity, in the MHz range, during the experiment. From these measurements it was possible to determine the freeze-thaw state of the soil, which was used for satellite imagery selection, and provide soil temperature measurements as modelling inputs to estimate the background scattering for the static sites.

At the centre of each snow survey site, a snowpit was excavated as a reference measurement. From the pit face, stratigraphy and layer heights were interpreted via visual inspection following standard methods (Fierz et al., 2009). Snow samples extracted from each layer were visually identified by grain origin type using a 2 mm comparator card and 40x magnification field microscope. Density profiles were collected from the pit face as continuous profiles with a Taylor–LaChapelle style cutter (100 cm $^3$; 3 cm height). Extracted samples of a known volume were immediately weighed with a shielded digital scale ($\pm 0.01$ g accuracy) to obtain density estimates. The consistent presence of vegetation voids and weak basal layers at the base of the pack made collection of continuous profiles challenging. As a result, measurements of basal hoar or ground-interfacing layers are likely to be underrepresented, a common issue in tundra studies (Domine et al., 2016). Profiles of snow specific area (SSA) were collected as an objective metric of microstructure with the A2 Photonics IceCube. The IceCube is a commercial implementation of infrared reflectometry sensors commonly used in snow studies (Domine et al., 2007; Gallet et al., 2009). Measurements of the reflected 1310 nm laser were calibrated using a set of 6 Spectralon diffuse reflectance targets before and after each profile. With a 3-cm extractor, samples were taken as continuous vertical profiles where cohesive samples could be extracted. For samples in depth hoar or vegetation voids, filling and packing of sampler was often required to ensure the laser would not fully penetrate the sample. Reported in $m^2kg^{-1}$, measurement uncertainty of SSA is expected to be $\pm 10\%$. See Figure A1 of Tsang et al. (2022) for a typical representation of vertical snowpit measurements.

At each snowpit site, a minimum of 2 Snow MicroPenetrometer (SMP; Proksch et al., 2015) profiles were collected near the central snow pit to calibrate against layered density and SSA from the typical snowpit measurements. The SMP measures the necessary force to drive the penetrometer at a consistent rate vertically through the snowpack (Schneebeli et al., 1999). SMP calibration measurements were taken within 10 cm of the snow pit face, adjacent to the profiles of density and SSA measurements. Calibration of the density and SSA models extends the work of King et al. (2020) and Calonne et al. (2020), as modified from the foundational work of Proksch et al. (2015) and Pielmeier and Schneebeli (2003). Following the pre-processing steps outlined in King et al. (2020), profiles penetrating less than 90% of the measured snow thickness were removed. Individual profiles were also evaluated to flag and remove signal artefacts including values below 0.001 N or changes in force of greater than 100% over distances less than 2.5 mm (Proksch et al., 2015; Marshall and Johnson, 2009). Applying a one-dimensional shot-noise process conceptualized in Löwe and van Herwijnen (2012), profiles of mean force ($\tilde{F}$) and length scale ($L$) of the penetration window, were computed for all SMP profile using a moving window of 5 mm with 50% overlap. The continuous profiles were then aggregated into 5 cm layers, small enough to represent average layers (Sandells et al., 2022). Results of the SMP calibration are shown in section 4.2. To capture vertical variability of snow properties within a 100 m footprint around the snowpit, a north-south and an east-west SMP transect was measured with a 10 m distance between each profile ($\sim 18$ profiles per site, Figure 3). These profiles were then converted to snow density and SSA using the calibration obtained from the pit profiles.

To get a better representation of snow depth distributions within the 100 m footprint around the snowpits, MagnaProbe (Sturm and Holmgren, 2018) measurements along the SMP transects were recorded every 1-2 meters ($\sim 290$ measurements per site, Figure 3).

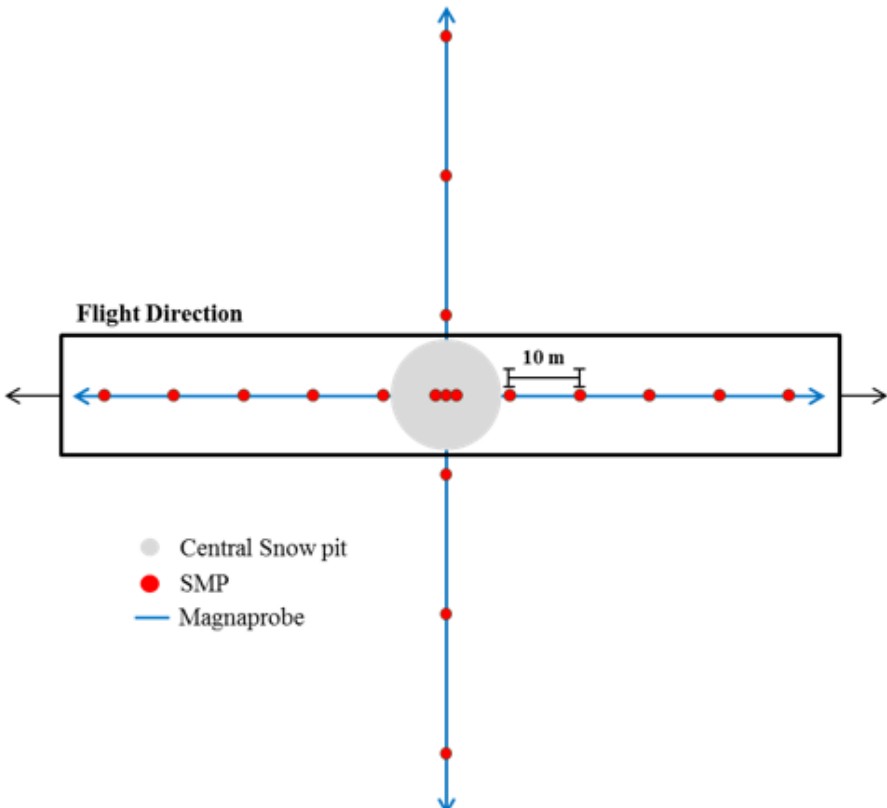

**Figure 3.** Ground based snow measurements sampling scheme. A complete snowpit is done at the center location of the surveyed site with three SMP profile measurements behind the snowpit wall. MagnaProbe (snow depth) and SMP transects are acquired in the North-South and East-West directions. MagnaProbe measurements are taken every meter and SMP measurements every 10 m.

In-addition to surveying the same 6 static sites during each campaign visit, snow measurements were also made at a series
of roving survey sites (i.e. Roving pits, RPs) in November (15), January (16), and March (25). Those sites were selected using a stratified random sampling approach which considered land cover and topographic variables (elevation, slope, orientation) sampled at the same spatial resolution and grid as the land cover data.. These extra sites were sampled in order to capture the variability of snow properties within the SikSik basin and how it impacts radar backscatter. They were also used to determine how representative the background soil properties measured at the static sites were of the entire domain by retrieving a distri-
bution of soil permittivity and roughness parameters from the forward modeling optimization process (section 3.3). The same sampling strategy was used for the static and roving sites for snow properties. Only the soil surface temperature was measured at the roving sites at the time of sampling.

## 3 Methods

This section provides the processing steps of the UMass radar data as well as the satellite imagery. The forward radiative transfer modelling optimization approach is also detailed.

### 3.1 Processing of SAR measurements

UMass radar measurements (2 x 2 m) were aggregated by considering pixels within a 50-m radius of central snow pits (n = 2500 pixels). The evaluated pixels were filtered to remove 3-sigma outliers and averaged to obtain a single value for analysis. This pre-processing was applied to minimize or negate the complex influence of pixel-scale variabilities (hummocks and vegetation) and radar speckle. Overlapping passes of the flight grid produced an average of 25 independent radar measurements per snow pit site (Table 2; 8 measurements in the worst case and 38 measurements for the best case). The overlapping flight lines provided diverse viewing geometry. On average, incidence angles ranging from approximately $19.5^o$ to $65.0^o$ were available at each site. The ability to view the same site from multiple incident-angles and look-direction configurations is a unique ability of airborne SAR, which in this study, enabled forward modeling and retrieval testing across a broad range of geometries.

The satellite imagery was processed using the ESA SNAP software and processing steps include: image calibration to $\sigma^0$; orthorectification (Range Doppler Terrain Correction) using the ArcticDEM (Porter et al., 2023) at 2 m spatial resolution; and extraction of the local incidence angle and its corresponding corrected backscatter values. An area of 100 x 100m with the snowpits geocoordinates as the centroids was extracted from the imagery. Approximately 1100 and 400 pixels were averaged for TSX ($\sim$3m pixel spacing) and RSAT-2 ($\sim$5m pixel spacing) respectively. No filtering was necessary given the wavelength of the two sensors and the fact that averaging over that large amount of pixels removes speckle noise.

### 3.2 Grain type classification

To represent the two-layer nature of the Canadian Arctic snowpack, the different layers of the SMP profiles were classified into rounded grains and depth hoar grains. To achieve this, the same support vector machine (SVM) classifier methodology developed in King et al. (2020) was used. To generate the SVM classifier, only the SMP profiles acquired behind the central snowpit wall were used as training data.

SMP measurements were used as input data to the classifier. Input variables consisted of: mean depth of the SMP snow layer, its associated median force ($\tilde{F}$) and length scale ($L$). The output label for each of the SMP layers were determined by the different snow layers visually identified by the surveyors at the snowpit wall. The surveyed grain type layer closest to the mean depth of the SMP layer was used as the output label. Some layers of mixed/faceted grains were identified by the surveyors, which do not directly correspond to the two dominant grain types. In order to use these layers, their labels were changed to rounded grains due to their visual similarity and consistency with what was reported by Picard et al. (2022), in terms of microwave grain size.

**Table 2.** Number of measurements per site for the TVC January 2019 campaign with its respective incidence angle range

| Site | Number of measurements | $\theta$ range |
|:---:|:---:|:---:|
| RP16 | 24 | 21.1 - 69.0 |
| RP17 | 38 | 18.8 - 64.9 |
| RP18 | 35 | 19.7 - 71.9 |
| RP19 | 37 | 17.9 - 70.2 |
| RP20 | 37 | 17.9 - 61.0 |
| RP21 | 28 | 18.9 - 69.8 |
| RP22 | 35 | 18.8 - 76.0 |
| RP23 | 19 | 19.3 - 64.9 |
| RP24 | 16 | 19.0 - 52.7 |
| RP25 | 22 | 17.5 - 65.3 |
| RP26 | 11 | 20.1 - 51.0 |
| RP27 | 12 | 17.7 - 63.6 |
| RP28 | 11 | 21.1 - 60.2 |
| RP29 | 13 | 18.2 - 61.0 |
| RP30 | 9 | 19.6 - 53.1 |
| RP31 | 8 | 23.3 - 76.1 |
| SC | 33 | 20.9 - 64.5 |
| SD | 37 | 19.7 - 70.9 |
| SM | 36 | 19.3 - 64.3 |
| SO | 31 | 18.8 - 62.5 |
| ST | 32 | 19.1 - 57.1 |
| SV | 35 | 22.4 - 80.3 |

## 3.3 Forward modelling

In this study, the Snow Microwave Radiative Transfer (SMRT, Picard et al., 2018) model was used to simulate the backscattered signal ($\sigma^0$) at C-, X-, and Ku-Band at VV polarization. SMRT is a multi-layered snow radiative transfer model where each layer is characterize by, minimally, its thickness, density, temperature, grain size (SSA, optical diameter or correlation length) and the model used to represent its microstructure. The calibrated SMP profiles provided thickness, correlation length and density and the temperature was inferred from the snowpit measurements. With these inputs, the microwave properties such as, interface reflectivity, volume scattering, and absorption are computed using the desired physical models, frequency and incidence angle. Finally, it solves the radiative transfer equation, to calculate the surface backscatter, in the case of active microwave sensors, using the Discrete Ordinate Radiative Transfer (DORT, Picard et al., 2004, 2013). To properly simulate $\sigma^0$, the following parameters need to be accurately estimated: 1. the background roughness and permittivity (Meloche et al.,

2021; Montpetit et al., 2018) and 2. the snow microwave grain size (Picard et al., 2022) related to microstructure and volume scattering. In this study, the Improved Born Approximation (IBA, Mätzler, 1998) was used for the volume scattering component with an exponential auto-correlation model to represent the snow microstructure, similarly to King et al. (2018); Montpetit et al. (2013).

For every optimization process at every site of the January 2019 campaign, the most representative SMP profile was selected to provide input of snow properties to SMRT for the multi-layered snowpack analysis. The number of layers were determined by the SMP profile processing described in section 2.3. The SMP profile selection was based on using the SMP profile with the snow depth that best corresponded to the median snow depth of all MagnaProbe measurements for a given site. For discussion purposes, and in the objective of improving computational efficiency, further testing using a two-layer snowpack was performed, where the median values of the rounded and depth hoar grain type layers, using all the measured data, including MagnaProbe, SMP profiles, and snowpits, was used to determine their snow geophysical properties (e.g., thickness, temperature, SSA, density).

For the background surface scattering modelling, the geometric optics (GO) model (Tsang and Kong, 2001), implemented in SMRT, was used since the surface roughness parameters at the 100-m scale, and the wavelengths of the sensors, largely surpasses the validity range of other model such as the Advanced Integral Equation Model (AIEM) (Meloche et al., 2021). The GO model is a high frequency approximation of the analytical Kirchhoff solutions, which describes the surface scattering of a very rough surface with no coherent scattering component. With little sensitivity to snow volume scattering of the signal intensity at C- and X-Band, the satellite data of RSAT-2 and TSX were used to retrieve the effective soil permittivity ($\varepsilon_{\text{soil}}$) for each band and effective soil roughness, i.e. mean square slope ($mss$), which is the parameter used in the GO surface scattering model.

Given the recent progress on understanding the microwave scattering properties of snow grains (Picard et al., 2022) and the two-layer nature of the Arctic snowpack (Rutter et al., 2019), it was decided to optimize the snow volume scattering using the polydispersity ($K$) parameter for two grain types, e.g. 1) rounded and 2) depth hoar grains. Those two grain types are the dominant grain in the two layer types reported by Derksen et al. (2012, 2009) for Canadian Arctic snowpacks. This means that the data at Ku-Band is used to retrieve three parameters: 1)$\varepsilon_{\text{soil}}[Ku]$, 2) rounded grain polydispersity ($K_{\text{R}}$), and 3) depth hoar polydispersity ($K_{\text{H}}$). In this study, polydispersity is simply retrieved, but such a parameter can be measured in the lab from micro-tomography measurements (Picard et al., 2022). This parameter describes the "non-uniformity" of the snow microstructure length scales in all directions. To describe the microwave snow grain size, polydispersity is a multiplying factor to the snow correlation length which can be estimated from SMP measurements.

With the amount of data available for all bands (Tables 1 and 2) and the diversity of SAR viewing geometry available, a simple least-squares method, using the Trust Region Reflective algorithm (Conn et al., 2009), was used to retrieve all parameters. This algorithm has the advantage of allowing boundary constraints which prevents the optimization process to converge on unrealistic values. The different effective parameters were thus constrained by values found in the literature.

The main objective of this study is to investigate the potential of SAR measurements to retrieve SWE.The fact that the signal intensity of lower frequencies (C- and X-band) is not sensitive to snow mass, these lower frequencies are first used to optimize

the soil background properties, without optimizing the snow volume scattering (i.e. polydispersity). The results from this first optimization will then be used to initialize and constrain the soil properties, and limit the number of variables to optimize in

the second optimization process using the Ku-band measurements. With this second optimization, the soil backscatter and the snow volume scattering will be considered.

## 4 Results

In this section, the seasonal evolution of snow properties as documented by the field measurements will be presented to provide context, even though only data from the January 2019 campaign will be used for the rest of the analysis and discussion. The

results of the SMP calibration with the snowpit measurements for both snow density and SSA will be shown. Retrieved background effective properties will be given as well as its error estimation for forward modelling at C- and X-Band. Finally, the forward modelling results at Ku-Band will be presented.

### 4.1 Spatio-temporal variability of snow properties

Figure 4 shows the seasonal evolution of snow depth, from MagnaProbe measurements, and depth hoar fraction, from classified

SMP profiles (Table 5), throughout the campaign. A median snow depth of 32 cm in the early season (November) was observed. Measurements show a median accumulation of up to 46 cm in the mid-season (January) and a median of 42 cm in the late season (March) with a more normal distribution across the entire domain. Figure 4 also shows that the fraction of the snowpack that consists of depth hoar is mainly concentrated around 0.5 during November and increases to 0.7 throughout the winter. There are fewer snow profiles that show no depth hoar from the classification as the winter season progresses.

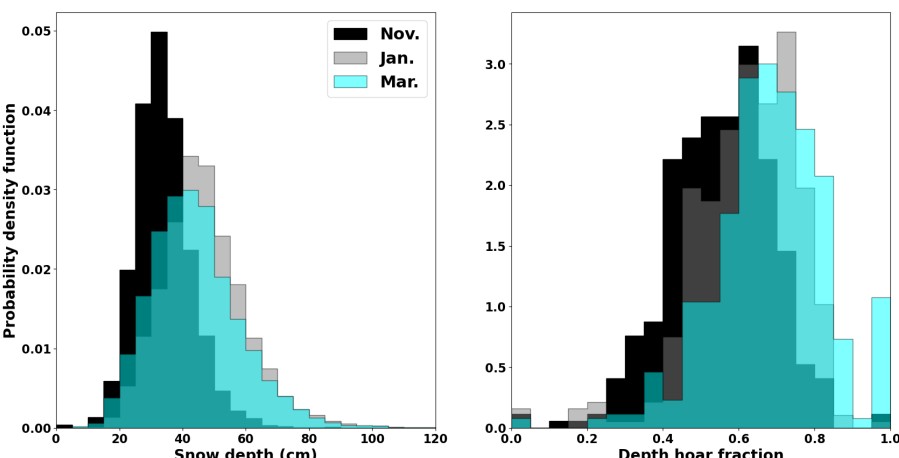

**Figure 4.** Evolution of snow depth distributions and depth hoar fraction throughout the field campaign.

Figure 5 shows the evolution of SSA, density, and temperature for the two dominant snow layers throughout the field campaign. Table 3 gives the median and standard deviations of the different snow properties measured for the entire campaign

shown in Figures 4 and 5. Figure 5 and Table 3 show that the grain size tends to get larger as the season progresses (i.e., lower SSA values). The only exception is during the January campaign where the rounded grain layer tends to have an increase in SSA followed by a decrease during the March campaign. As for density, an overall densification of the snowpack for both snow layers was observed and a slight decrease in density can be seen from January to March. The snow temperatures reflect the air temperature trends for each site visit, where colder temperatures and warmer air temperatures were measured during the January and March campaigns respectively compared to the November campaign.

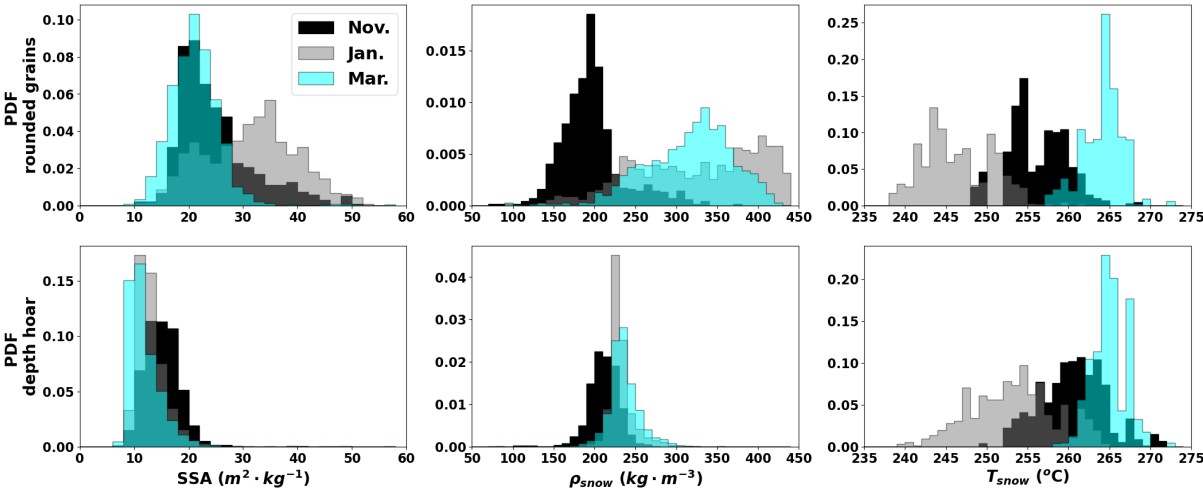

**Figure 5.** Evolution of snow geophysical properties (SSA, density, and temperature) throughout the winter season for the two dominant snow grain type layers: rounded (top row) and depth hoar grains (bottom row).

**Table 3.** Median and standard deviations of measured snowpack properties during the TVC 2018/19 winter season for the two dominant snow grain type layers: rounded (R) and depth hoar grains (H).

| Site | Grain Type | Nov. | Jan. | Mar. |
|---|---|---|---|---|
| Depth (cm) | — | 33.2 (9.5) | 45.6 (13.2) | 42.5 (14.0) |
| H fraction | — | 0.6 (0.1) | 0.6 (0.1) | 0.7 (0.2) |
| SSA $(m^2 \cdot kg^{-1})$ | R | 21.9 (7.0) | 31.7 (8.7) | 20.7 (4.8) |
| | H | 14.5 (4.1) | 12.4 (3.5) | 11.0 (4.0) |
| $\rho_{snow}$ $(kg \cdot m^3)$ | R | 193.7 (35.1) | 334.0 (80.1) | 320.5 (56.1) |
| | H | 212.3 (21.4) | 229.2 (20.2) | 233.2 (23.0) |
| $T_{snow}$ $(^oC)$ | R | -16.9 (3.4) | -27.8 (4.2) | -8.8 (2.6) |
| | H | -12.1 (3.9) | -19.9 (5.1) | -8.2 (2.0) |

## 4.2 Characterization of snow properties

Figures 6 and 7 show the results of the density and SSA estimates from the calibrated SMP measurements. Table 4 gives the
equations used to calibrate the SMP profiles to $\rho_{snow}$ and SSA.

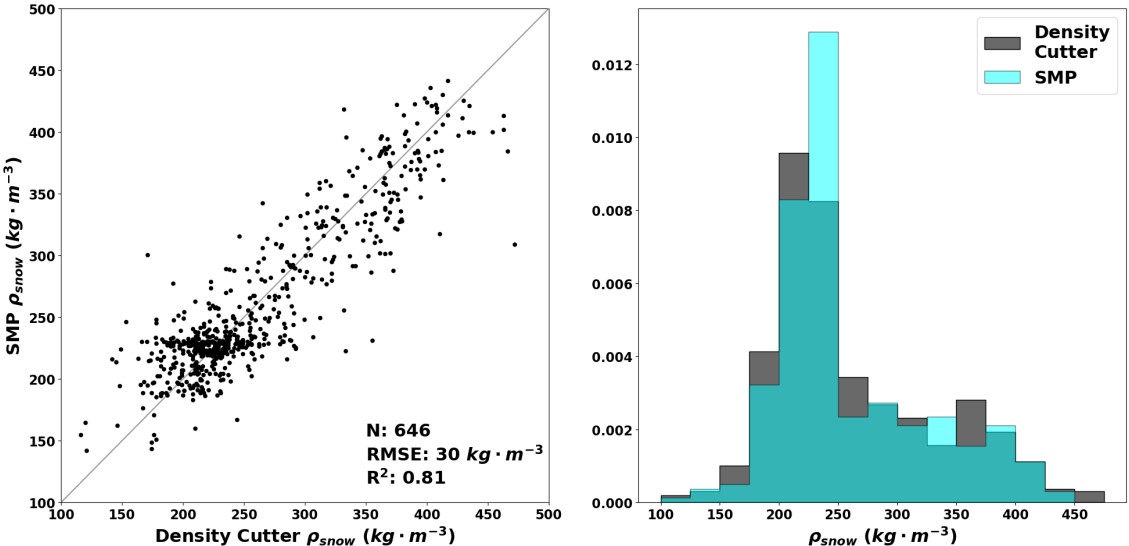

**Figure 6.** Results of calibrated SMP snow density measurements. Left panel shows the comparison of the calibrated SMP density measurements with the density measurements taken with a density cutter. Right panel compares the distributions of both measurements, i.e. density cutter and SMP densities.

Following the methodology of King et al. (2020), good agreement is achieved between the SMP and density cutter measurements. With 646 comparison points, we get an RMSE = 31 $kg \cdot m^{-3}$ (12% error) and an $R^2$ = 0.81, which is comparable to results obtained by King et al. (2020) for snow on sea ice and to the results of (Dutch et al., 2022) who used a subset of these same field measurements while studying the impact of snow properties on heat transfer within the snowpack. Figure 6 also
shows that the density measurement distributions for both the SMP and density cutter overlap well, which further validates the measurement agreement.

Figure 7 shows that the calibration coefficients of Calonne et al. (2020) do not generate SSA values comparable to IceCube measurements for low SSA values (< 15 $m^2 \cdot kg^{-1}$). Given this result, new coefficients were generated for this study (King-TVC). With 627 comparison points, we get an RMSE = 5.2 $m^2 \cdot kg^{-1}$ (29% error) and $R^2$ = 0.68 with this new calibration
compared to RMSE = 6.0 $m^2 \cdot kg^{-1}$ (35% error) and $R^2$ = 0.57 with the Calonne et al. (2020) calibration. The systematic underestimation of low SSA values from the SMP measurements is also removed with the new calibration.

Figure 8 illustrates an example of the distribution of MagnaProbe snow depths, the corresponding snow depth for the different SMP measurements, and the selected profile which is closest to the median value of the MagnaProbe distribution for a given site.

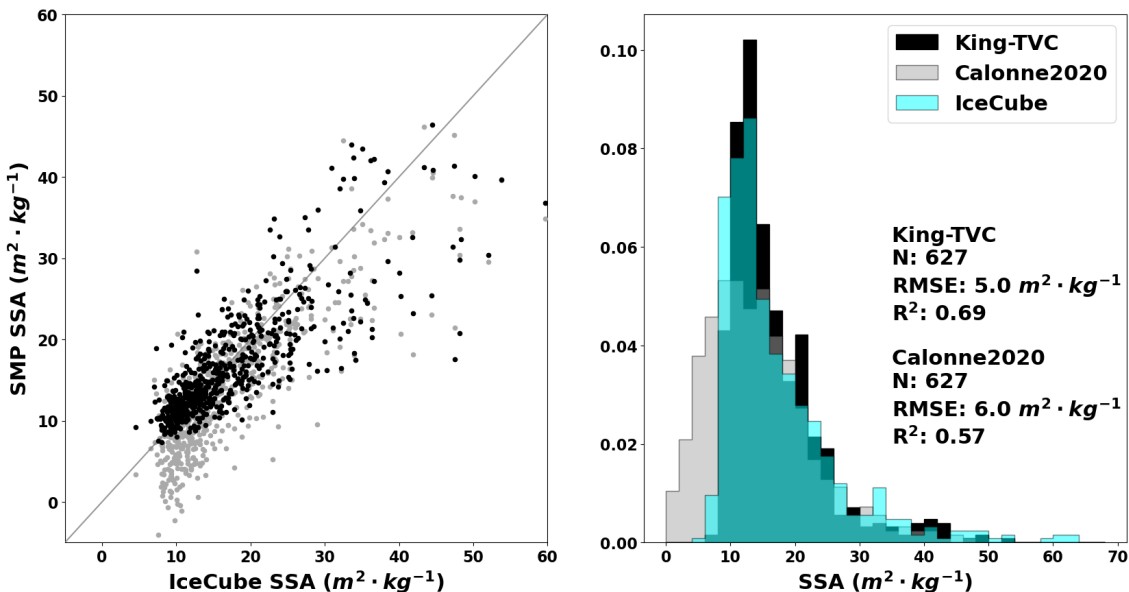

**Figure 7.** Results of calibrated SMP SSA measurements. Left panel shows the comparison between the calibrated SMP SSA with the IceCube measurements, using two different calibration: Calonne et al. (2020), gray dots, and the proposed calibration of table 4, black dots. Right panel compares the SSA distributions of both SMP calibrations and the IceCube measurements.

**Table 4.** Density and SSA calibration equations used for Figures 6 and 7.

| Calibration | Equation |
|---|---|
| Calonne et al. (2020) | $SSA = 0.57 - 18.56 \ln(L) - 3.66 \ln(\tilde{F})$ |
| King-TVC | $\rho_{\text{snow}} = 307.76 + 53.81 \ln(\tilde{F}) - 44.24 \ln(\tilde{F})L - 64.8L$ <br> $SSA = \exp(2.37 - 0.70 \ln(L) - 0.06 \ln(\tilde{F}))$ |

In order to retrieve the polydispersity parameters to optimize the snow volume scattering at Ku-Band, the different snow layers were classified into rounded grains versus depth hoar layers using the SVM classifier described in section 3.2. The similar $\rho_{\text{snow}}$ and $SSA$ distributions (Figure 9) of the mixed/faceted layers with the rounded grain layers confirms the decision to convert the prior type as rounded grains.

     Table 5 shows the confusion matrix of the classification results. An overall accuracy of 88.4% ($\pm 2.5\%$) was obtained over
10 different randomly shuffled iterations with an 80/20% split for training and testing, respectively.

     With this classifier, 67% of the layers that were classified as mixed/facets layers by the surveyors were classified as rounded grains and the other 33% as depth hoar.

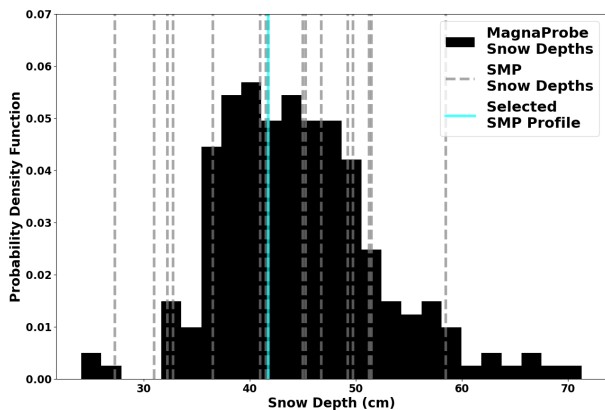

**Figure 8.** Example of the SMP profile selection as a representative snowpack for the SM site (Table 2). The histogram represents the distribution of snow depth measured with the MagnaProbe. The vertical lines represent the measured snow depth of the different SMP profiles.

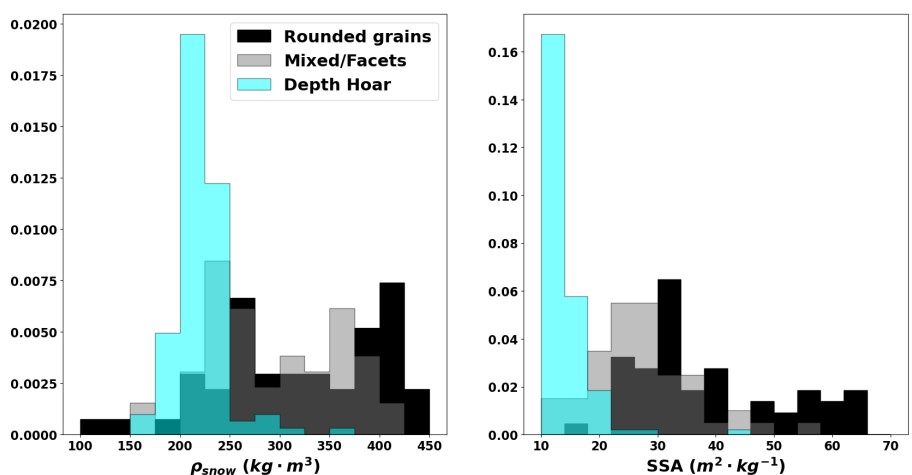

**Figure 9.** Distribution of snow density and SSA for the three dominant grain type layers for the January campaign.

## 4.3 Forward modelling of C- and X-Band backscatter

This section presents the forward modelling optimization of the background soil properties at C- and X-Band, using the January
campaign data alone to focus solely on dry winter conditions. Initial values of permittivity for the optimization process were extrapolated from the retrieved values of Montpetit et al. (2018). The boundaries for which the optimization process could not go beyond were determined by published values ($2.3 \leq \varepsilon_{\text{soil}} \leq 5.0$, Meloche et al., 2021; Pulliainen et al., 1999). The initial mean square slope, i.e. soil roughness ($mss_{\text{soil}}$) value was set by the median value obtained from airborne LiDAR measurements collected in August 2018 before the field campaign ($mss_{\text{soil}}$=0.011, Lange et al., 2021). The range was determined by the

**Table 5.** Confusion matrix of the grain type classification, using the SVM classifier (section 3.2), for the January snowpits.

|  |  | Predicted | |
|--|--|--|--|
|  |  | R | H |
| Obs. | R | 0.83 | 0.17 |
|  | H | 0.18 | 0.82 |

standard deviation of these measurements ($\pm 0.010$). Since the LiDAR point clouds were noisy with inconsistent point density for all the sites of the January campaign, these measurements were not used directly to simulate the backscatter at the different frequencies. For all parameters, none of them had converged towards the upper or lower boundaries for any sites meaning there was always a minima within the constrained values. Table 6 shows the results of the background optimization for all January sites (including static and roving sites) and the static sites individually to show the variability in the different land coverages

of the SikSik basin. Out of the six static sites, only the one site (SC02, table 2) was neglected for the parameter retrieval due to the fact that a permanently installed radar corner reflector was mounted nearby within the 100 m footprint surrounding the static snow measurement site. This corner reflector artificially increased the backscatter at C- and X-Band for this site and was thus not considered for the retrieval.

**Table 6.** Retrieved soil parameters using C- and X-Band data for the static sites individually and all surveyed sites including the static sites (all) of the January campaign.

| Site | $\varepsilon_{\text{soil}}$ | | $mss_{\text{soil}}$ |
|--|--|--|--|
|  | C-Band | X-Band |  |
| All | 2.47+i0.0045 | 2.61+i0.0061 | 0.010 |
| SD | 2.63+i0.0026 | 2.95+i0.0051 | 0.011 |
| SM | 2.32+i0.0018 | 2.44+i0.0025 | 0.011 |
| SO | 2.54+i0.0033 | 2.40+i0.0022 | 0.010 |
| ST | 2.27+i0.0017 | 2.38+i0.0021 | 0.009 |
| SV | 2.50+i0.0032 | 2.50+i0.0028 | 0.010 |

   Figure 10 show the distributions of all the retrieved real $\varepsilon_{\text{soil}}$ values for all January sites. Both frequencies have very similar

distributions with slightly different median values (Table 6). Overall, we get $\varepsilon_{\text{soil}} = 2.47 + i0.0045(\pm 0.21 + i0.0014)$ and $\varepsilon_{\text{soil}} = 2.61 + i0.0061(\pm 0.23 + i0.0012)$ for C- and X-Band, respectively. Uncertainties were calculated using the standard deviation of the retrieved parameters for all January sites. As shown in Figure 10, both frequencies have similar uncertainties. The global mean square slope roughness parameter was $mss_{\text{soil}} = 0.010(\pm 0.002)$. Given that the $mss_{\text{soil}}$ value is centered at 0.010 with very little variability, this validates the use of a single value for all sites.

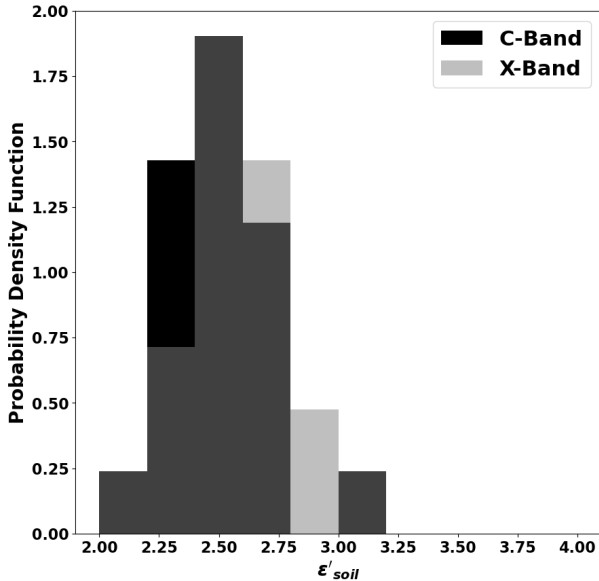

**Figure 10.** Distributions of the retrieved real $\varepsilon_{\mathrm{soil}}$ for C- and X-Band data for all the sites measured during the January campaign.

Figure 11 displays the results between the measured and simulated backscatter values for all snow survey sites of the January campaign for both bands. Figure 11 a) shows the results when simulating the backscatter with a single set of parameters for the entire domain and Figure 11 b) for simulated backscatter using retrieved values for each site individually (distributed values of Figure 10).

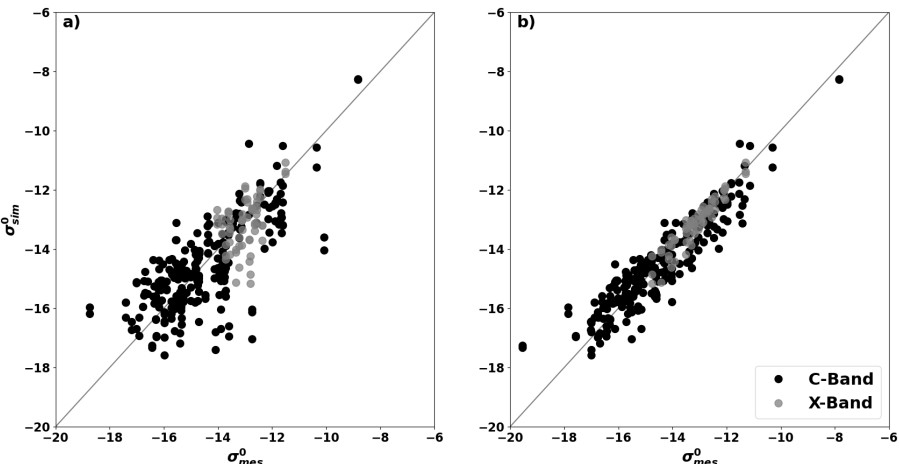

**Figure 11.** Comparison between simulated and measured $\sigma^0$ at C- and X-Band for a) using a single set of parameters for all sites ('All' in Table 6) and b) retrieved parameters for each site individually (distributed values shown in Figure 10)

Overall, there is a larger spread in Figure 11 a), which translates into larger errors (RMSE=1.1 dB and bias=0.1 dB; RMSE=0.7 dB and bias=0.0 dB for Figures 11 a) and b), respectively).

## 4.4 Forward modelling of Ku-Band backscatter

Since Ku-Band data is sensitive to snow volume scattering, polydispersity parameters for the two dominant grain types had to be considered in this optimization process. As shown in Section 4.3, the soil permittivity in the GO surface scattering model is frequency dependent. Since the $mss_{\text{soil}}$ is considered independent of frequency, a single parameter was used for all sites ("All" in Table 6). This optimization process thus had four parameters to optimize in total. For the initial permittivity value, the parameter was set by extrapolating from the two previous values retrieved in section 4.3. The same boundaries were set as the optimization at C- and X-Band. The scaling factor $\phi = 1.09$, which is comparable to polydispersity (see section 4.4), obtained by King et al. (2018) for TVC depth hoar dominated snowpacks was used as the initial optimization polydispersity value. A slightly wider range of values, than published by Picard et al. (2022), was used to constrain the range of plausible values, i.e. 0.5 to 1 for rounded grains and 1 to 2 for depth hoar compared to 0.6 to 0.9 for rounded grains and 1.2 to 1.9 for Canadian Arctic depth hoar.

Table 7 illustrates the median and standard deviation values of the retrieved parameters over the January sites. Figure 12 shows the distributions of the retrieved parameters.

**Table 7.** Retrieved soil and snow parameters using Ku-Band data for January.

| $\varepsilon_{\text{soil}}$ | $K_{\text{R}}$ | $K_{\text{H}}$ |
|---|---|---|
| 2.77+i0.7406 (0.75+i0.15) | 0.74 (0.15) | 1.11 (0.26) |

Compared to Figure 10, there is a much larger spread in distribution of the real part of the $\varepsilon_{\text{soil}}$ at Ku-Band and seems to have two clusters, one centered at 2.41 and the other at 3.82.

For the polydispersity values ($K$), the values retrieved for the rounded grains ($K_{\text{R}}$) show a distribution centered around 0.74. The values for the depth hoar grains ($K_{\text{H}}$) show a different spread but most values are centered around 1.11.

Figure 13 a) shows the results when simulating the backscatter with a single set of parameters for all the domain and Figure 13 b) for simulated backscatter using retrieved values for each site individually (distributed values of Figure 12).

Similarly to Figure 11 there is less spread and better agreement between the simulated and measured $\sigma^0$ when considering distributed parameterizations instead of a single set of parameters for the entire domain. Mean bias and RMSE of 0.9 dB and 2.6 dB for Figure 13 a) and, -0.1 dB and 0.9 dB for Figure 13 b) respectively. Figure 13 c) shows the same figure as Figure 13 a) but with median values of the two different $\varepsilon'_{soil}$ clusters. These results show less spread and better accuracy (bias = 0.0 dB and RMSE = 1.3 dB) than a single set of permittivity values for all sites.

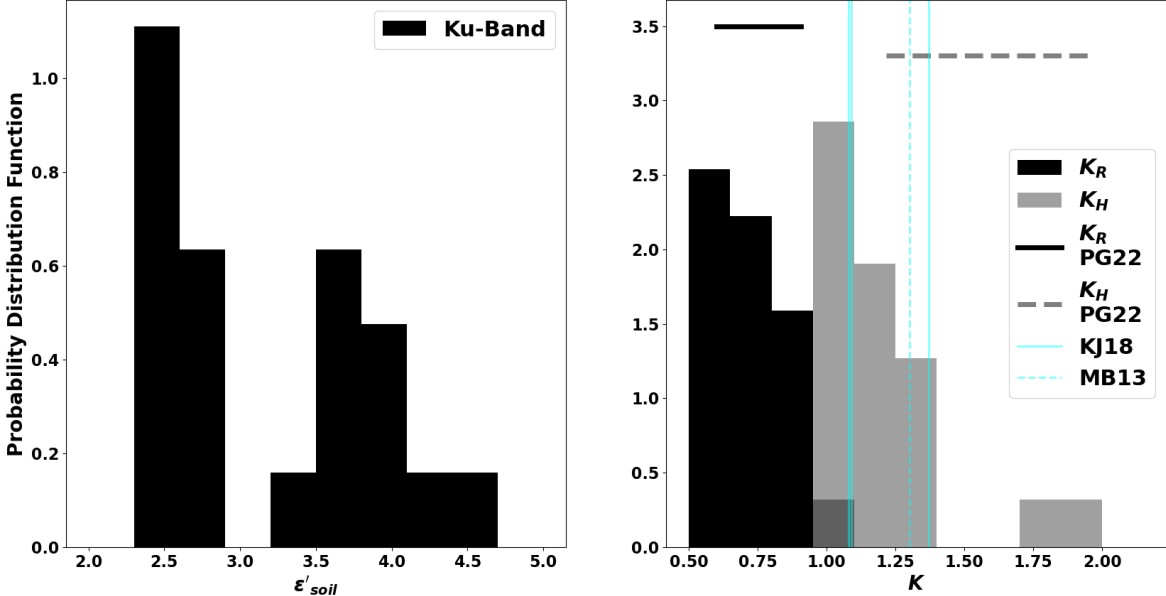

**Figure 12.** Distribution of retrieved parameters at Ku-Band (Table 7). The range of retrieved $K$ values (horizontal lines at the top of the plot) from Picard et al. (2022) for both grain types (PG22) and the different values retrieved by King et al. (2018) and Montpetit et al. (2013) (vertical lines) are also displayed (KJ18 and MB13 respectively).

## 5 Discussion

### 5.1 Characterization of snow properties for radar

SMP measurements have become increasingly useful to better characterize spatial variability of the snowpack properties in the field (King et al., 2020; Hagenmuller and Pilloix, 2016; Teich et al., 2019; Tsang et al., 2022). As reported by King et al. (2020), no single method works for every SMP instrument in every study area even though there are different published calibration parameters and methods (Proksch et al., 2015; King et al., 2020; Pielmeier and Schneebeli, 2003). This means that SMP calibration against density and SSA measurements is required for each instrument and field campaign. In this study we show that the approach of King et al. (2020) can retrieve snow density and SSA from SMP profiles rapidly and efficiently given the proper snowpit sampling strategy (section 2.3). For SSA, low values when using the previous work of Calonne et al. (2020) were improved by 6% overall and 17% for low SSA values by generating new calibration coefficients for the SMP data. In order to increase sensitivity to the low SSA values, the equation of Table 4 were fitted against the log of the reference SSA values. This change in calibration equation did not have a significant impact on the RMSE, i.e. from 5.3 $m^2 \cdot kg^{-1}$ to 5.2 $m^2 \cdot kg^{-1}$) when using the linear versus log scale SSA values, respectively. A significant improvement was observed on the $R^2$ values, passing from 0.24 to 0.68 when using linear and log scale SSA values respectively.

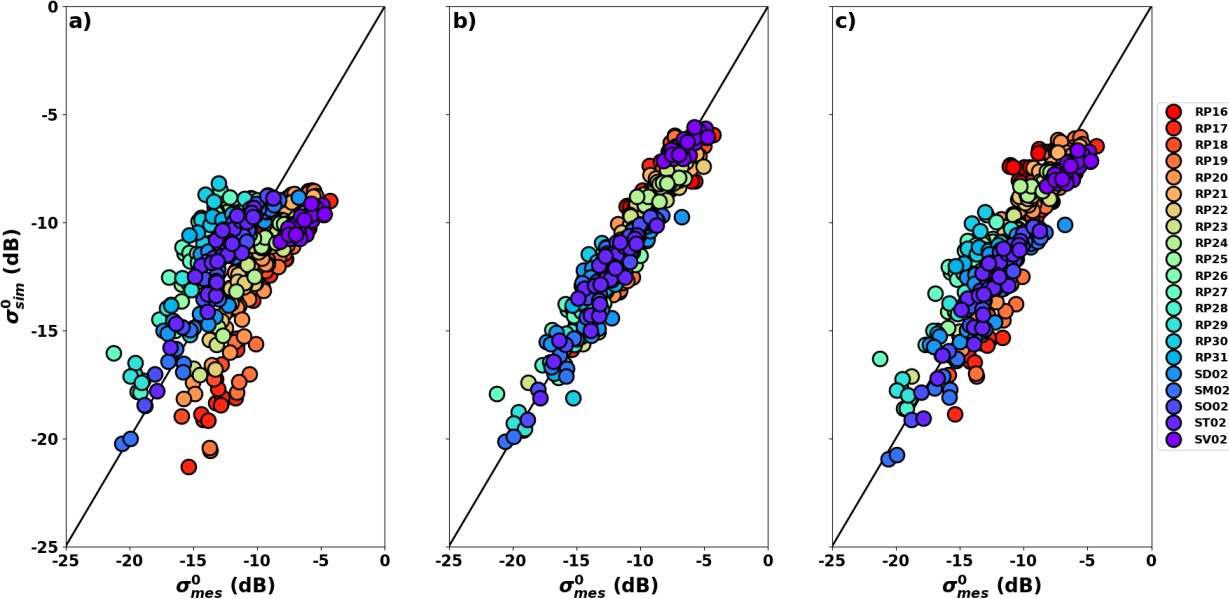

**Figure 13.** Comparison between simulated and measured $\sigma^0$ at Ku-Band for a) using a single set of parameters for all sites (Table 7), b) retrieved parameters for each site individually (distributed values shown in Figure 12) and c) the same parameterization as a) except the median values of $\varepsilon'_{soil}$ of the two clusters of Figure 12 were used. Color code corresponds to each surveyed site in January.

Snow layers of the SMP profiles, were classified into the two dominant snow type categories for Canadian Arctic (Derksen et al., 2009; Picard et al., 2022), to simplify the generation of statistically representative snowpacks. It should be noted that a fresh snow layer was not considered in this study since it was not present during the January campaign. Having 33% of the mixed/faceted layers reported by surveyors for the January campaign classified as depth hoar can be explained by its overlapping SSA and $\rho_{snow}$ distributions (Figure 9) with the two main grain types. Faceted grains found during the 2018/19 winter season at TVC consists of faceted rounded particles as reported in Picard et al. (2022) for Antarctica and alpine snowpacks, which is why those layers were originally labelled as rounded grains for the classification. The similar distributions between the mixed/faceted grains and the rounded grains also support this assumption. Some solid faceted particles were identified in the mixed/faceted layers surveyed, which might explain the lower $SSA$ distribution compared to rounded grains (Figure 9). This could also explain why 33% of the faceted crystals are linked to depth hoar.

The overall evolution of snow depth and depth hoar fraction (Figure 5) observed throughout the winter season is similar to what was previously reported in studies for TVC (King et al., 2018; Dutch et al., 2022). The SSA and $\rho_{snow}$ distributions during the March campaign might be underestimated. This is due to the difficult surveying conditions due to air temperatures above 0 $^o$C on some survey dates. Warm air temperatures make it difficult to retrieve proper samples for IceCube and density measurements, especially for depth hoar layers as snow sticks to the instruments. Some liquid water content was also present in some snow samples which impacts density and SSA measurements. All these uncertainties result in less accurate calibration

of the SMP data. Unfortunately, with the liquid water content potentially present in the snowpack during some radar flights, it will be challenging to validate these snow geophysical properties against Ku-Band radar data.

## 5.2 Forward modeling of C- and X-Band backscatter

The fact that with two frequencies, the $mss_{\mathrm{soil}}$ values all converge towards a single value, i.e. $mss_{\mathrm{soil}} = 0.010$, with little variability (18%) indicates that this value can be used at the satellite/airborne scale for radiative transfer modelling. This value
is also identical to the $mss_{\mathrm{soil}} = 0.01$ reported by Zhu (2021). Even though, the LiDAR data was not used directly for the backscatter simulations, the retrieved value is in good agreement with the measured median value of 0.011 obtained over the January sites after filtering the extreme values due to noise and anthropogenic sources.

As shown by Tsang and Kong (2001), the GO model is not frequency dependent, but results indicate that this is due to the permittivity value used in the model. The low variability (7%) observed in the retrieved permittivity values for both C-
and X-Band and the small errors between the simulated and measured backscatter, indicates that the intensity signal at these frequencies is not impacted by snow volume scattering for Arctic tundra snowpacks found at TVC. Also, the small variability in permittivities indicate that the ground signal is fairly stable, which suggests the signal penetrates into the soil surface and is less impacted by the variable surface vegetation composition.

## 5.3 Forward modeling of Ku-Band backscatter

Figure 12 shows a large spread of background soil permittivities which indicates that the Ku-Band signal is much more sensitive to the composition of the soil surface than the other two lower frequencies analyzed in this study. In fact, there are two distinct permittivity clusters. The land cover types at the survey sites associated with the lower value cluster are mostly dominated by lichen and tussocks, whereas the land cover types at the sites with the higher values are mostly dominated by lichen and dwarf shrubs. The results here are in agreement with permittivity values reported by Savin and Mironov (2020) where higher
permittivities were found for sites dominated by shrubs compared to sites dominated by tussocks. Since the retrieved effective permittivities encompasses both soil and vegetation, it is possible that the higher permittivity values compensate for higher scattering from the shrubs.

With only three data points, it is difficult to extrapolate the effective permittivity values to higher frequencies using the GO model. Zhu (2021) reported that permittivities should saturate in the Ku- to K-Band frequency range using the same background
model. This implies that little variability should be observed for permittivities at higher frequencies than at Ku-Band in this study. Results of tables 6 and 7 show and increase of the soil permittivity with frequency. This goes against the modelled frequency dependency of soil permittivity (Mironov et al., 2017; Zhang et al., 2010), which highly depends on the permittivity of ice and the ice-fraction in the soil. Montpetit et al. (2018) reported the same frequency dependency as shown here, from retrieved permittivity values using passive microwave radiometer data, and a different reflectivity model for similar soil types.
This tends to indicate that retrieved values from microwave measurements (both active and passive) are sensitive to different components of the soil vertical profile, where models describe the frequency dependence of homogeneous soil samples.

The fact that the optimization process shows sensitivity to grain size via the polydispersity ($K$) values for both grain types (Figure 12b) are in agreement with the values reported by Picard et al. (2022). Results show that the lower portion of the Ku-Band spectrum is sensitive to snow volume scattering. The $K$ values retrieved for both dominant grain types (Figure 12b) are in agreement with the values reported by Picard et al. (2022). Results show that the exponential correlation length parameter, used for snow grain size, has to be reduced for rounded grain layers and boosted for depth hoar layers in order to increase and reduce the snow volume scattering for their respective layers. The $K_R$ values are normally distributed around 0.74 and range between 0.5 and 1, as established by Mätzler (2002). The values for $K_H$ do not really follow a specific distribution and show a wider spread, indicating larger uncertainty on the depth hoar polydispersity. Though no significant relationship was found, the higher values of $K_H$ tend to be associated with higher depth hoar fraction ($> 0.45$). The median value retrieved of 1.11 is also in agreement with grain size correction factors ($\phi$), which can now be explained by the polydispersity (Picard et al., 2022), reported by King et al. (2018) and Montpetit et al. (2013) for Canadian Arctic tundra sites. Those studies applied a single correction factor to all layers and it is known that the microwave snow volume scattering is dominated by the depth hoar layer which tends to boost the overall polydispersity close to $K_H$ in this case. While not investigated in depth in this study, using a single polydispersity value for both grain types resulted in $K = 1.5$ which further supports the dominance of the volume scattering component by the depth hoar layer.

The larger spread and lack of agreement between simulated and measured backscatter values in Figure 13 a) can be explained by the model not taking into account the two different clusters of soil permittivity observed in Figure 12. When using the median values from the two different clusters there is improved agreement and reduced bias as confirmed by Figure 13 c). In fact, changing the polydispersity values within the retrieved range did not have a significant impact on the overall error (<0.5 dB improvement of RMSE). This indicates that the lower spectrum of the Ku-Band is still sensitive to the background surface scattering even in the presence of snow volume scattering (which was negligible at C- and X-Bands). With the saturation effect of the modelled background properties within the Ku-Band range, the volume scattering will only become more dominant compared to the background surface scattering. These results show that a distributed, statistical approach, for all the retrieved parameters, is more suited to forward modelling of the Ku-Band signal (Pan et al., 2023), even though this approach is less preferable for satellite observation SWE retrievals (Durand et al., 2024). This is mainly due to the high computation cost of statistical approaches. To improve efficiency in forward modelling computation time, the snowpacks of January were reduced to two layers, i.e. rounded grains and depth hoar layers, where the median value of all the measured data, including MagnaProbe, SMP, and snowpits, was considered to generate the geophysical properties of both layers. No significant change was observed in the RMSE ($\sim 7\%$ difference) between the simulated and observed backscatter values. This further supports the two-layer classification approach used in this study and confirms that two layers are sufficient to represent a Canadian tundra snowpack (Derksen et al., 2012; King et al., 2018).

## 6    Conclusions

This study describes in detail the spatio-temporal evolution of snow geophysical properties during the Trail Valley Creek experiment conducted during the winter of 2018/19. It was shown that the Snow Micro Penetrometer is an efficient instrument

to quickly and quantitatively determine spatial variability of the vertical snow structure within a given footprint, representative of a single grid cell measurement at satellite scale. It was also demonstrated that the Canadian Arctic tundra snowpack is well represented by a two layer snowpack consisting of a wind-compacted rounded grain layer and a depth hoar layer.

The main objective of this study is to investigate the potential of SAR data to retrieve snow water equivalent (SWE). It is well documented that Ku-band SAR is well suited to retrieve SWE but underlying soil backscatter has to be properly estimated. This was achieved using satellite data from RADARSAT-2 and TerraSAR-X, the background soil contribution to measured backscatter was characterized. An RMSE of 0.7 dB was achieved between the simulated and measured backscatter at C- and X-Band using the retrieved background properties. Using the Geometrical Optics surface scattering model, we found that the real part of the effective permittivity tends to increase with frequency. The retrieved parameters were then used to constrain the optimization of soil backscatter and snow volume scattering at Ku-band.

Following the constraint of the soil background properties, the contribution of snow volume scattering at Ku-band was also optimized. Using the two-layer classification approach for all the different layers measured by the SMP, we showed that the snow volume scattering was dominated by the depth hoar layer, where $K_H$ increased the grain size, thus its volume scattering ($\sim 1.11$), and the $K_R$ of the rounded grain layer reduced it ($\sim 0.74$). An overall RMSE between the simulated and measured backscatter at Ku-Band of 0.9 dB was achieved when using the distributed retrieved values of soil permittivity and snow polydispersity. This confirms that a statistical approach is better suited to reproduce the measured radar backscatter from ground geophysical properties (Pan et al., 2023) rather than using a single set of values to represent a larger domain such as TVC.

This validates the use of the SMRT model and its different subroutines, i.e. Geometrical Optics for soil surface backscatter, Improved Born Approximation (IBA) with an exponential autocorrelation function for snow scattering and the Discrete Ordinate Radiative Transfer (DORT) solver, to forward simulate the airborne UMASS radar measurements. These results confirm the development direction of the snow water equivalent retrieval algorithm for the future Canadian Terrestrial Snow Mass Mission (TSMM), which will use SMRT simulations.

The fact that the background properties saturate in the Ku-Band spectrum further validates the proposed use of the dual-frequency Ku-Band (13.5 and 17.25 GHz) TSMM concept. The lower Ku frequency is more sensitive to the soil backscatter contribution than the higher Ku frequency, due to the higher sensitivity to snow volume scattering at the higher frequency. The fact that the background properties should be similar for both frequencies, due to the saturation of the permittivity in the Ku range (Zhu, 2021), would allow isolation of the background surface scattering component from the snow volume scattering component of the signal received by a dual-frequency sensor.

Having the two frequencies will also allow for the better estimation of depth hoar fraction using the retrieved polydispersity values from this study. Having a first guess of the snow vertical profiles from a land surface model like the Soil Vegetation Snow Version 2 (SVS-2 Garnaud et al., 2019; Vionnet et al., 2022), using Crocus as the snow model component (Vionnet et al., 2012), should further constrain the plausible snow physical properties within a known distributed range of values which will allow the measured Ku-Band radar backscatter to be related to the bulk SWE values using the SMRT scheme presented in this study.

In order to improve computational efficiency, future work needs to be conducted in order to reduce the number of snow
layers of the land surface models to a number of layers relevant to radar radiative transfer modelling, i.e. a "microwave relevant"
snowpack. We have shown that for the snowpack measured at TVC, reducing the number of layers to the two main snow grain
types, i.e. depth hoar and rounded grains, is appropriate for skillful forward modelling of the radar signal in the tundra region.

*Data availability.* The TerraSAR-X data are available through the DLR (©DLR 2019). RADARSAT-2 Data and Products ©MacDonald,
Dettwiler and Associates Ltd. (2018) – All Rights Reserved. RADARSAT is an official trademark of the Canadian Space Agency.

*Code and data availability.* All codes are available at https://github.com/ECCCBen/TVCExp18-19. Links to the different datasets used will
also be provided in the github repository.

*Author contributions.* BM, JK and CD wrote the manuscript with contributions from all co-authors. JK designed the experiment. PS, MA
and his team at UMass developed the airborne radar and processed the data. BM, JK and JM performed the analysis. BM, JK, CD and
PT collected the field measurements. MB helped write portions of code used and reviewed the codes before publication. AW ordered and
provided the TSX data. VV and NL reviewed the manuscript and provided analysis guidance in the context of the TSMM mission.

*Competing interests.* Some authors are members of the editorial board of journal The Cryosphere.

*Acknowledgements.* This work was started and field campaign orchestrated by the late Joshua King. The study was completed by the other co-
authors. Trail Valley Creek activities were supported by Environment and Climate Change Canada, the Canadian Space Agency and NASA's
THP and ESTO-IIP programs (Grant numbers: 80NSSC20K1592, 80NSSC22K0279). The authors would like to thank the excellent logistical
support provided by the Trail Valley Creek station crew, in particular Branden Walker and Philip Marsh. This work would not have been
possible without the contribution of many partners including Barum Majumber (WLU), Alexandre Roy (UQTR), Alex Mavrovic (UQTR),
Daniel Kramer (UdS), Simon Levasseur (UdS), Casey Wolieffer (UMass), Nick Rutter (Northumbria U.), Richard Essery (Northumbria U.),
Jim Hudgson (Lake Central Aircraft Services), Yves Crevier (CSA) and Simon Yueh (NASA).

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
