# Peer review of "Retrieval of snow and soil properties for forward radiative transfer modeling of airborne Ku-Band SAR to estimate snow water equivalent: The Trail Valley Creek 2018/19 Snow Experiment"

_EGUsphere, 2024_

## Referee Comment (RC1)

General Comments

The authors of the manuscript describe the capability of backscatter retrieval using snow microstructure measurements as inputs in a radiative transfer model, SRTM. The paper presents a viable case for a dedicated spaceborne snow mission and explains how the backscatter measurements can be used to characterize snowpack properties. The paper combines methodologies used in previous studies for classification and optimization with novel snowpack measurement techniques to estimate background and snow volumetric backscatter.

The paper is well thought out and constructed, with appropriate references for each step. However, the section describing the radiative transfer model itself could be modified to include more information regarding the model. Some of the methods appear in the results section for the first time, which can confuse readers. Adding a flowchart showing the complete methodology could make it easier to follow.

The results are encouraging, with a low RMSE of 0.9 dB, especially for Ku Band. The paper demonstrates publication quality in terms of conceptualization and execution of the study, but a more detailed description of some of the methodology could be provided.

Line 12: can be "characterized"..

Line 13: I believe this result is important. A sensitivity analysis of soil background roughness on backscatter for different frequencies can be provided. This implies that for a particular snow regime, this value can be used as an initial guess with hard constraints to reduce the number of parameters and simplify the optimization studies done in the future.

Line 25-27: Perhaps it's a case of a misplaced comma, but the statement lacks parallel structure. The first part refers to coarse resolution SWE "products," whereas the second part refers to high spatial resolution "sources."

Line 32: Multiple studies demonstrate the viability of C-band for retrieving wet snow pixels. Perhaps a brief discussion could be added on how Ku-band improves our retrieval capabilities compared to our previous estimates.

Line 158: As SMP measurements are the basis of this study a small paragraph on the working principle of the instrument can be added.

Line 178: How is the stratification done using the combination of a categorical (Land Cover) and continuous variable (topography)?

Line 193: Word "ranging from" can be added for clarity.

Line 204: For improved clarity for those unfamiliar with the model, a concise overview of SMRT along with its input parameters can be provided.

Line 208: The constraints should be discussed along with references in a table for reproducibility.

Line 213 and Section 4.3: The actual effect of snowpack on the high frequency backscatter should be discussed. It is not clear how the effect of snow volume backscatter and ground backscatter were separated.

Line 269: It is a clever way to identify rounded grains and depth hoar layers. A brief description of SVM classification methodology can be provided in the methods section. Additionally, a brief description of the training datasets for grain type identification is important information for various kinds of studies.

Line 282: Why wasn't the mean square slope calculated directly using the Lidar point cloud?

Line 381: Figure no. should be mentioned in the bracket, even though it is mentioned in the starting of the section.

Line 399: Why a distributed, statistical approach is not preferable for SWE retrievals using satellite observations? The results in the reference are not based on scatterometer data.

Figure 2: The study area figures can be improved. The fonts in the legends are small and difficult to read. Maybe the figure

Figure 5: The histograms are slightly difficult to interpret in the overlapping areas. Therefore, if possible, the histograms can be replaced with lines for better interpretability.

Figure 8: I do not understand what is being shown in the figure. Does the dashed line represent the depths where the SMP measurements were made?

 Figure 13: The legend can be provided outside the figure and scaled up for clarity.

.

---

## Author Comment (AC1)

**Reply to comments from Reviewer 2**

Montpetit et al.

**Correspondence:** Benoit Montpetit (benoit.montpetit@ec.gc.ca)

**1 General Comments**

The study presents an experiment simulating radar backscattering intensity of snow-covered ground in a tundra environment using a coupled ground-snow radiative transfer model. Model simulations, driven by a combination of pre-retrieved and measured parameters, are compared to backscattering at Ku-band measured by an airborne radar instrument. Ground properties

5    defining surface backscattering are retrieved from satellite-based measurements at lower frequencies, using these to guide the Ku-band retrievals. The forward simulation setup is used to retrieve optimized values for snow polydispersity in a two-layer simulation setting.

The text is generally well written and clear, and the subject is of high interest to the community since it provides new insight into the recently proposed concept of microwave grain size and how this is related to radar backscatter observed in remote sensing.

10   However, I feel the section describing the forward model setup is currently insufficient to fully comprehend the experiment, with some details explained only in the discussion. The origin and use of structural polydispersity, a key aspect of the paper, is not fully explained, nor how this is used together with measured optical grain size to calculate the scattering coefficient in radiative transfer. Furthermore, some measurements such as those on soil permittivity, are meticulously presented but finally not used anywhere in the paper. Likewise, it is unclear how snow pit measurements were finally used in the simulations. These

15   aspects should be improved before considering the paper ready for publication. Please see detailed comments below.

Figures are clear and present the results in a useful way, although figure captions are on occasion too concise to fully describe the figure contents. This could be improved.

The authors would like to thank the reviewer for the positive, constructive and thorough review. All of the major/minor comments have been address below. Along with comments from reviewer #1, they significantly improved the quality of the

20   manuscript. In particular, the methodology section has been improved to provide more details on the radiative transfer model used and its parameterization as well as the details on the SVM model used to classify grain type.

**2   Major comments**

Section 3.2 is the main problem of the paper, and the only reason why I suggest a major revision. Many key details are missing, and the text in the section does not fully describe the model setting. E.g., which of the multitude of SMRT scattering models was used? How many layers were used in the simulation? How were other parameters than the retrieved ones defined for these layers (e.g. snow depth and density)? One has to read between the lines or reach out all the way to the end of the discussion to get some answers. For example, line 208 refers to "the multi-layer analysis", but it is not clear how many layers there are (two, three, or more?). This only becomes clear (perhaps) only later in the manuscript, that two layers were used. As a further example, the last sentence states that "The different effective parameters were thus constrained by values found in the literature", without giving the values. Please modify the entire section and explain fully 1) the model setup, including which scattering model was used in SMRT 2) state carefully all model parameters and from what source these were derived from (measured, retrieved, literature etc). Maybe a Table could help? E.g. at present it is not fully traceable how all the measurements described in Section 2.3 were finally used in simulations, validation or both.

This section has been revised with more details on the SMRT setup, the multi-layer (which corresponds to the results of Figure 13) and two-layer approach taken and which dataset (MagnaProbe, SMP, snowpit, soil) was used as input to SMRT.

*In this study, the Snow Microwave Radiative Transfer (SMRT, Picard et al., 2018) model was used to simulate the backscattered signal ($\sigma^0$) at C-, X-, and Ku-Band at VV polarization. SMRT is a multi-layered snow radiative transfer model where each layer is characterize by, minimally, its thickness, density, temperature, grain size (SSA, optical diameter or correlation length) and the model used to represent its microstructure. The calibrated SMP profiles provided thickness, correlation length and density and the temperature was inferred from the snowpit measurements. With these inputs, the microwave properties such as, interface reflectivity, volume scattering, absorption are computed using the desired physical models, frequency and incidence angle. Finally, it solves the radiative transfer equation, to calculate the surface backscatter, in the case of active microwave sensors, using the Discrete Ordinate Radiative Transfer (DORT, Picard et al., 2004, 2013). Of the inputs, to properly simulate $\sigma^0$, the following parameters need to be accurately estimated: 1. the background roughness and permittivity (Meloche et al., 2021; Montpetit et al., 2018) and 2. the snow microwave grain size (Picard et al., 2022) related to microstructure and volume scattering. In this study, the Improved Born Approximation (IBA, Mätzler, 1998) was used for the volume scattering component with an exponential auto-correlation model to represent the snow microstructure, similarly to King et al. (2018); Montpetit et al. (2013). [...]*

*The number of layers were determined by the SMP profile processing described in section 2.3. The SMP profile selection was based on using the SMP profile with the snow depth that best corresponded to the median snow depth of all MagnaProbe measurements for a given site. For discussion purposes, and in the objective of improving computational efficiency, further testing using a two-layer snowpack was performed, where the median values of the rounded and depth hoar grain type layers, using all the measured data, including MagnaProbe, SMP profiles, and snowpits, was used to determine their snow geophysical properties (e.g., thickness, temperature, SSA, density).*

The section could also benefit from a (brief) introduction of the SMRT and the GO models. Just a sentence or two placing the models in context for a potential reader who has no idea what these models actually do, could be sufficient. e.g. "SMRT (Picard et al., 2018) simulates propagation of microwaves in snow, generating estimates of microwave emission and backscatter from a stacked system of snow layers, with each layer described by..." and so on.

The two models are now introduced in section (see previous comment for SMRT):

*The GO model is a high frequency approximation of the analytical Kirchhoff solutions, which describes the surface scattering of a very rough surface with no coherent scattering component.*

A similar short intro should be added on snow microstructure, in particular the relatively new concept of microwave grain size introduced by Picard et al., how this is obtained from field measurements, and where Polydispersity comes in. Maybe this warrants a separate subsection in Methods?

We feel an entire subsection on polydispersity is not needed because it would repeat what is described in depth in Picard et al. (2022). Instead, we have added a few lines on polydispersity to Section 3.3:

*In this study, polydispersity is simply retrieved, but such a parameter can be measured in the lab from micro-tomography measurements (Picard et al., 2022). This parameter describes the "non-uniformity" of the snow microstructure length scales in all directions. To describe the microwave snow grain size, polydispersity is a multiplying factor to the snow correlation length which can be estimated from SMP measurements.*

**3 Minor comments**

The title is a bit awkward to me. "Retrieval of airborne Ku-band SAR..." does not really say anything (retrieval of what? radar backscatter?). Please consider changing the title to e.g. "Retrieval of snow and ground properties from airborne Ku-band SAR...", or even "Retrieval of snow polydispersity and ground effective permittivity from Ku-band SAR..." since those are the parameters you finally retrieve.

Good point. The title was changed to:

*Retrieval of snow and soil properties for forward radiative transfer modeling of airborne Ku-Band SAR to estimate snow water equivalent: The Trail Valley Creek 2018/19 Snow Experiment*

Abstract line 4. Not sure about "quality snow information". I would change this to something less ambiguous, e.g. simply "snow information"

Has been changed to *accurate snow information* since the objective is to indicate that these measurements don't simply provide snow information but information of a certain "value".

Abstract line 6. "It becomes possible to properly characterize" is also quite a strong statement. What is properly? I'd suggest something like "(SMP and SMRT)... are promising tools for characterizing the snow cover ... etc"

85    *Text has been changed to:*

*Recently developed tools like the Snow MicroPenetrometer (SMP), to retrieve snow microstructure data in the field, and radiative transfer models like the Snow Microwave Radiative Transfer Model (SMRT), are promising tools for characterizing snow and how it translate into radar backscatter.*

Abstract line 15. Polydispersity is a very new concept in snow microwave modeling and you should give some short context
90    for the term. e.g. "The polydispersity of the snow microstructure, which together with snow autocorrelation length has been proposed as a basis for scattering calculations of microwaves propagating in snow,..." (please use some wording better fitting your text). See also later comment on adding a paragraph, or even a separate section, on the concept of microwave grain size.

*Text has been modified to:*

*Recently, it was shown that snow grain size, represented by exponential correlation length, could be translated to its equivalent*
95    *"snow microwave grain size" by a parameter called polydispersity. Values of 0.74 and 1.11 for rounded and depth hoar snow grains polydispersity, respectively, was retrieved.*

Abstract, last sentence. Since you have not demonstrated actual SWE retrieval in this study, I suggest to modify this a bit: "... using a priori knowledge of the snow conditions to simulate backscattering, facilitating also the possible retrieval of SWE from these measurements" (or similar).

100    *Text has been changed to:*

*[...] , which is of great importance in SWE retrieval algorithms from Ku-Band SAR measurements.*

Introduction line 38-39: "decomposing..." I'd rather say the overall problem in SWE retrieval is separating the influence of both variable microstructure and ground backscatter on total backscatter (as opposed to separating these from one another, maybe just change "from" to "and"?). Also change "ground permittivity" to "ground backscatter". I also suggest to change the
105    wording so that you acknowledge these vary also spatially, not only temporally.

*Text has been modified to include suggestions:*

*Advancement of a Ku-band radar-based SWE retrieval is highly dependent on decoupling the strong spatial and seasonal influences of snow microstructure and background backscatter (Picard et al., 2022; Meloche et al., 2021).*

Intro line 68. "Is most sensitive to SWE when a priori snow microstructure is known" does not really make sense. The sensitivity
110    to SWE remains the same (weak or strong) whether or not we know what the microstructure is. I guess you mean "the sensitivity to SWE can be predicted when a priori snow microstructure is known", or similar. Please reword.

Reworded:

*These studies (e.g., Tsang et al., 2022; Lemmetyinen et al., 2022; King et al., 2018) have shown that the Ku-Band frequency range is most sensitive to SWE, and a priori knowledge of snow microstructure is necessary to accurately estimate SWE.*

Section 2.1 line 117 "prevents good forward modeling" is a bit inaccurate, since you could simulate also wet snow effects with e.g. SMRT, which would be just as "good". Rather, the problem is that meaningful retrievals of SWE, microstructure or ground properties become difficult due to the overwhelming influence of wet snow on the radar signal? Please reword.

Reworded:

*Some liquid water content in the top portion of the snowpack was also present which reduces the influence of ground properties and snow microstructure in forward modelling of the snowpack due to the overwhelming influence of wet snow on the radar signal.*

section 2.2 lines 128-131. The sentence is a bit awkward to me. Maybe split into two sentences after "were considered"?

Sentence was split and second sentence was reworded to:

*[...] . Only imagery that showed intensity variability (standard deviation) below 2 dB, from one image to another, for all surveyed sites, were considered.*

section 2.3 lines 142: you state that "it was possible to determine the freeze-thaw state of the soil and provide modelling inputs to estimate the background scattering". However, to what I see, these measurements were finally not used, but all ground values were retrieved ones? Did you try comparing retrievals to measured permittivities? If not, it is not necessary to report these measurements.

The permittivity and roughness were only retrieved yes. The retrieved permittivities were not compared with the measured ones since they were not measured for the same frequency range. Nonetheless, they were used to ensure that the soil state was constant in the 2-weeks buffer period used for satellite imagery selection. The soil temperature was used as a model input for the static sites, though it had little influence on the soil backscatter since the permittivity was retrieved. The dataset is kept in the manuscript since it is linked to the rest of the campaign and the dataset is published with the rest. The text was modified to clarify:

*The soil sensor networks collected hourly measurements of temperature, moisture, and complex permittivity, in the MHz range, during the experiment. From these measurements it was possible to determine the freeze-thaw state of the soil, which was used for satellite imagery selection, and provide soil temperature measurements as modelling inputs to estimate the background scattering for the static sites.*

Section 4.1 line 240. The values stated in the text seem to contradict Figure 4 and Table 3, please verify.

Modified:

*[...] mainly concentrated around 0.5 during November and increases to 0.7 throughout the winter.*

Line 259 typo, remove (Calonne2000)

Removed

145 Section 4.4 line 307. Permittivity parameter? Do you mean soil permittivity?

Yes, it was changed in the text:
*As shown in Section 4.3, the soil permittivity in the GO surface scattering model is frequency dependent.*

lines 311-312. Here you introduce a previous scaling factor, but no explanation is given why this should be comparable to polydispersity. One has to dig into the paper by Picard et al. for clues, but it should be explained it here. See previous comment
150 on adding short introduction/explanation of the microwave grain size concept.

The concept is now introduced in the methods section. Text was modified here as well:
*The scaling factor $\phi = 1.09$, which is comparable to polydispersity (see section 4.4), [...]*

Figure captions in general, here Figure 6 as an example: The captions should enable the reader to understand the figure without referring to the text. Instead of a generic "Results of calibrated SMP snow density measurements", explain the two panels in
155 the figure (scatterplot and histogram) fully in the caption, including information on what was used in the calibration (i.e. snow pit profiles). Please check that captions of all figures and Tables provide sufficient information.

Captions have been modified to be self sufficient.

line 281 typo: Grandell

Great catch, the downloaded bibtex file from Scopus.com was wrong. New file from IEEE was used to generate the proper
160 citation/reference to (Pulliainen et al., 1999)

Discussion: did you compare retrieved soil permittivity values at all to measurements? What does literature and models say about the frequency dependence of permittivity of frozen ground (largely controlled by permittivity of ice)? At least the latter should be discussed, even though the retrievals represent "effective" values. If a comparison to the measured ground permittivities did not yield anything conclusive, this could also be at least mentioned.

165 A comparison with measured soil permittivities was not conducted since the measurements were made in the MHz range and cannot be compared with retrieved permittivities in the GHz range. Text was added to discuss the frequency dependence of frozen ground permittivity:
*Results of tables 6 and 7 show and increase of the soil permittivity with frequency. This goes against the modelled frequency*

*dependency of soil permittivity (Mironov et al., 2017; Zhang et al., 2010), which highly depends on the permittivity of ice and the ice-fraction in the soil. Montpetit et al. (2018) reported the same frequency dependency as shown here, from retrieved permittivity values using passive microwave radiometer data, and a different reflectivity model for similar soil types. This tends to indicate that retrieved values from microwave measurements (both active and passive) are sensitive to different components of the soil vertical profile, where models describe the frequency dependence of homogeneous soil samples.*

Discussion, lines 400-403. One has to read all the way here to understand the simulation setup. Please move this explanation to Methods, adding other necessary details (see previous comments)

Some of this information was included in section 3.3 but the following text, in section 3.3 was modified to improve the simulation setup description. This is in addition to the description of the SMRT model and its functionalities used in this study. *For every optimization process at every site of the January 2019 campaign, the most representative SMP profile was selected to provide input of snow properties to SMRT for the multi-layered snowpack analysis. The number of layers were determined by the SMP profile processing described in section 2.3. The SMP profile selection was based on using the SMP profile with the snow depth that best corresponded to the median snow depth of all MagnaProbe measurements for a given site. For discussion purposes, and in the objective of improving computation efficiency, further testing using a two-layer snowpack was performed, where the median values of the rounded and depth hoar grain type layers, using all the measured data, including MagnaProbe, SMP profiles, and snowpits, was used to determine their snow geophysical properties (e.g., thickness, temperature, SSA, density).*

Conclusions, lines 418-419. Increased and decreased compared to what? I am assuming the ranges introduced by Picard et al., but this should be made clear. Also, note that Picard et al presented the first estimates of these parameters, so finding values outside of these ranges is not surprising.

Reworded:

*[...] we showed that the snow volume scattering was dominated by the depth hoar layer, where $K_H$ increased the grain size, thus its volume scattering ($\sim 1.11$), and the $K_R$ of the rounded grain layer reduced it ($\sim 0.74$).*

line 430 "background properties should be similar..." I do not follow. Why should these be similar?

Reworded with regards to the conclusions of Zhu (2021) who stated that the soil backscatter should saturate in the Ku-K-Band range.
*The lower Ku frequency is more sensitive to the soil backscatter contribution than the higher Ku frequency, due to the higher sensitivity to snow volume scattering at the higher frequency. The fact that the background properties should be similar for both frequencies, due to the saturation of the permittivity in the Ku range (Zhu, 2021), would allow isolation of the background surface scattering component from the snow volume scattering component of the signal received by a dual-frequency sensor.*

line 442: "...skillful forward modelling of the radar signal in the tundra region".

200    Modified.

**References**

[revised manuscript text omitted]

---

## Author Comment (AC2)

**Reply to comments from Reviewer 1**

Montpetit et al.

**Correspondence:** Benoit Montpetit (benoit.montpetit@ec.gc.ca)

**1 General Comments**

The authors of the manuscript describe the capability of backscatter retrieval using snow microstructure measurements as inputs in a radiative transfer model, SRTM. The paper presents a viable case for a dedicated spaceborne snow mission and explains how the backscatter measurements can be used to characterize snowpack properties. The paper combines methodologies used in previous studies for classification and optimization with novel snowpack measurement techniques to estimate background and snow volumetric backscatter. The paper is well thought out and constructed, with appropriate references for each step. However, the section describing the radiative transfer model itself could be modified to include more information regarding the model. Some of the methods appear in the results section for the first time, which can confuse readers. Adding a flowchart showing the complete methodology could make it easier to follow. The results are encouraging, with a low RMSE of 0.9 dB, especially for Ku Band. The paper demonstrates publication quality in terms of conceptualization and execution of the study, but a more detailed description of some of the methodology could be provided.

The authors would like to thank the reviewer for the great feedback provided which considerably improves the manuscript. By answering the specific comments below, the methodology section has been improved with details on the radiative transfer model used and the support vector machine classifier used to classify grain types.

**2 Specific comments**

Line 12: can be "characterized"..

Corrected.

Line 13: I believe this result is important. A sensitivity analysis of soil background roughness on backscatter for different frequencies can be provided. This implies that for a particular snow regime, this value can be used as an initial guess with hard constraints to reduce the number of parameters and simplify the optimization studies done in the future.

We agree that this is an important result of this study. Since it was conducted on a single study area, it is difficult to extrapolate this result to other areas that differ in land cover types. Also, via the optimization process, some sensitivity to background roughness was evaluated and is reflected in the standard deviation of the retrieved value. Though this study uses multi-frequency SAR, the focus of this study was to retrieve snow parameters at Ku-Band, thus adding a multi-frequency sensitivity analysis to the paper would be out of scope for this study and would lengthen it. That said, this analysis will be done in the context of the Terrestrial Snow Mass Mission algorithm development and will be published in a report or possibly peer-reviewed paper/letter. To clarify, the following text was added to section 3.3:

*The main objective of this study is to investigate the potential of SAR measurements to retrieve SWE.The fact that the signal intensity of lower frequencies (C- and X-band) is not sensitive to snow mass, these lower frequencies are first used to optimize the soil background properties, without optimizing the snow volume scattering (i.e. polydispersity). The results from this first optimization will then be used to initialize and constrain the soil properties, and limit the number of variables to optimize in the second optimization process using the Ku-band measurements. With this second optimization, the soil backscatter and the snow volume scattering will be considered.*

It was also reiterated in the conclusions:

*The main objective of this study is to investigate the potential of SAR data to retrieve snow water equivalent (SWE). It is well documented that Ku-band SAR is well suited to retrieve SWE but underlying soil backscatter has to be properly estimated. This was achieved using satellite data from RADARSAT-2 and TerraSAR-X, the background soil contribution to measured backscatter was characterized. An RMSE of 0.7 dB was achieved between the simulated and measured backscatter at C- and X-Band using the retrieved background properties. Using the Geometrical Optics surface scattering model, we found that the real part of the effective permittivity tends to increase with frequency. The retrieved parameters were then used to constrain the optimization of soil backscatter and snow volume scattering at Ku-band.*

Line 25-27: Perhaps it's a case of a misplaced comma, but the statement lacks parallel structure. The first part refers to coarse resolution SWE "products," whereas the second part refers to high spatial resolution "sources."

Text has been modified to:

*While surface snow depth observation networks support the generation and validation of coarse resolution (>25 km) snow water equivalent (SWE) products from passive microwave remote sensing (e.g., Luojus et al., 2021), higher spatial resolution (<500m) SWE products are needed to meet the needs of climate services, water resource management, and environmental prediction (Garnaud et al., 2019, 2021; Kim et al., 2021; Cho et al., 2023).*

Line 32: Multiple studies demonstrate the viability of C-band for retrieving wet snow pixels. Perhaps a brief discussion could be added on how Ku-band improves our retrieval capabilities compared to our previous estimates.

It is true that C-Band is a viable option to retrieve wet snow conditions. The physics behind the detection of wet snow at both C- and Ku-Band are practically the same. That said, C-Band can be sensitive to deeper liquid water content (percolation) where

the signal at Ku-Band might saturate due to its lower penetration depth within the snowpack. We agree that both frequencies are complementary to detect wet snow conditions. Text has been modified:

*[...] (2) the ability to discriminate wet from dry snow cover (Tsang et al., 2022), as a complement to existing C-Band SAR methods (Stiles and Ulaby, 1980).*

Line 158: As SMP measurements are the basis of this study a small paragraph on the working principle of the instrument can be added.

A sentence was added to describe the working principal of the instrument. The rest of the paragraph explains how the force measurements of the penetrometer are calibrated for snow microstructure measurements:

*The SMP measures the necessary force to drive the penetrometer at a consistent rate vertically through the snowpack (Schneebeli et al., 1999).*

Line 178: How is the stratification done using the combination of a categorical (Land Cover) and continuous variable (topography)?

Topographic elements (height, orientation, slope) were mapped to the same resolution as the land cover data and the stratified random sampling was done from these combined variables. Modification to text:

*[...] using a stratified random sampling approach which considered land cover and topographic variables (elevation, slope, orientation) sampled at the same spatial resolution and grid as the land cover data.*

Line 193: Word "ranging from" can be added for clarity.

Sentence was changed to:

*On average, incidence angles ranging from approximately 19.5º to 65.0º were available at each site*

Line 204: For improved clarity for those unfamiliar with the model, a concise overview of SMRT along with its input parameters can be provided.

Included SMRT overview at the beginning of section 3.3 (Methods/Forward Modeling):

*In this study, the Snow Microwave Radiative Transfer (SMRT, Picard et al., 2018) model was used to simulate the backscattered signal ($\sigma^0$) at C-, X-, and Ku-Band at VV polarization. SMRT is a multi-layered snow radiative transfer model where each layer is characterize by, minimally, its thickness, density, temperature, grain size (SSA, optical diameter or correlation length) and the model used to represent its microstructure. In this study, the calibrated SMP profiles provided thickness, correlation length and density and the temperature was inferred from the snowpit measurements. With these inputs, the microwave properties such as, interface reflectivity, volume scattering, and absorption are computed using the desired physical models, frequency and incidence angle. Finally, it solves the radiative transfer equation, to calculate the surface backscatter, in the case of active microwave sensors, using the Discrete Ordinate Radiative Transfer (DORT, Picard et al., 2004, 2013). To properly simulate*

*$\sigma^0$, the following parameters need to be accurately estimated: 1. the background roughness and permittivity (Meloche et al., 2021; Montpetit et al., 2018) and 2. the snow microwave grain size (Picard et al., 2022) related to microstructure and volume scattering. In this study, the Improved Born Approximation (IBA, Mätzler, 1998) was used for the volume scattering component with an exponential auto-correlation model to represent the snow microstructure, similarly to King et al. (2018); Montpetit et al. (2013).*

Line 208: The constraints should be discussed along with references in a table for reproducibility.

Did the reviewer meant line 228 with regards to the boundaries of the optimized soil/snow values. The snow polydispersity constraints were already presented in section 4.4 with its reference (Picard et al., 2022), but it is true that the values for the $\varepsilon_{soil}$ and $mss_{soil}$ were not. These latter values are now added in section 4.3 with their relevant references:
*($2.3 \leq \varepsilon_{\text{soil}} \leq 5.0$, Meloche et al., 2021; Pulliainen et al., 1999).*
A table was not added since all values are already included in the text. These values are also discussed in sections 5.2 and 5.3. Finally, for reproducibility, the codes and data are all available here https://github.com/ECCCBen/TVCExp18-19, where these values are integrated within the Notebooks of Parts 7 and 8.

Line 213 and Section 4.3: The actual effect of snowpack on the high frequency backscatter should be discussed. It is not clear how the effect of snow volume backscatter and ground backscatter were separated.

Within the context of radiative transfer, unfortunately, the snow volume scattering and the ground surface backscatter cannot be decoupled since the Discrete Ordinate Radiative Transfer (DORT, Picard et al., 2004, 2013) solver requires both components together, in order to simulate the backscatter at the surface of the snowpack. The effect of snow volume scattering is discussed in section 5.3 where the snow volume scattering is driven by the depth hoar layer. The fact that the two grain types are sensitive to the polydispersity values is proof of the sensitivity of the low Ku-Band signal to snow volume scattering. This is now explicitly stated in section 5.3:
*The fact that the optimization process shows sensitivity to grain size via the polydispersity ($K$) values for both grain types indicates that the lower portion of the Ku-Band spectrum is sensitive to snow volume scattering. [...] Results show that the exponential correlation length parameter, used for snow grain size, has to be reduced for rounded grain layers and boosted for depth hoar layers in order to increase and reduce the snow volume scattering for their respective layers.*

Line 269: It is a clever way to identify rounded grains and depth hoar layers. A brief description of SVM classification methodology can be provided in the methods section. Additionally, a brief description of the training datasets for grain type identification is important information for various kinds of studies.

A new section was added to the methods section to briefly describe the grain type classifier.
*To represent the two-layer nature of the Arctic snowpack, the different layers of the SMP profiles were classified into rounded grains and depth hoar grains. To achieve this, the same support vector machine (SVM) classifier methodology developed in*

*King et al. (2020) was used. To generate the SVM classifier, only the SMP profiles acquired behind the central snowpit wall*
115  *were used as training data.*

*SMP measurements were used as input data to the classifier. Input variables consisted of: mean depth of the SMP snow layer, its associated median force ($\tilde{F}$) and length scale (L). The output label for each of the SMP layers were determined by the different snow layers visually identified by the surveyors at the snowpit wall. The surveyed grain type layer closest to the mean depth of the SMP layer was used as the output label. Some layers of mixed/faceted grains were identified by the surveyors, which do not*
120  *directly correspond to the two dominant grain types. In order to use these layers, their labels were changed to rounded grains due to their visual similarity and consistency with what was reported by Picard et al. (2022), in terms of microwave grain size.*

Line 282: Why wasn't the mean square slope calculated directly using the Lidar point cloud?

This was initially tested. Unfortunately, as mentioned in section 4.3 line 284, the LiDAR point cloud is very noisy and impacted by anthropogenic sources as well as having inconsistent point density for the different sites. This generated extreme
125  values that were unrealistic. Nonetheless, as stated, the median value for the entire study area was used as the initial conditions. Statistically, the median value filtered out the extreme values.

Line 381: Figure no. should be mentioned in the bracket, even though it is mentioned in the starting of the section.

Added

Line 399: Why a distributed, statistical approach is not preferable for SWE retrievals using satellite observations? The results
130  in the reference are not based on scatterometer data.

The retrieval approach in (Pan et al., 2023; Durand et al., 2024) uses the same radar SnowSCAT scatterometer data. One uses a Bayesian approach (Pan et al., 2023) and the other uses a look up table (Durand et al., 2024) to retrieve SWE from the SnowSCAT data. In both cases, they used radiative transfer models similar to the one used in this study either directly in their retrieval or to generate their look up table.
135  The reason why the statistical approach is less desirable for satellite retrieval of SWE is due to the computation cost. This has been added in the text:
*This is mainly due to the high computation cost of statistical approaches.*

Figure 2: The study area figures can be improved. The fonts in the legends are small and difficult to read. Maybe the figure

Figure size has been increased to fit the full width of the page.

140  Figure 5: The histograms are slightly difficult to interpret in the overlapping areas. Therefore, if possible, the histograms can be replaced with lines for better interpretability.

Figure 5 has been modified to improve interpretability. The vertical lines of the bars have been removed, which improves the visibility of histogram overlaps.

Figure 8: I do not understand what is being shown in the figure. Does the dashed line represent the depths where the SMP
145 measurements were made?

The dash lines represent the depths of the different measured SMP profiles and the histogram shows the spread of all snow depth measurements using the MagnaProbe for a given site. These details have been added to the figure caption.
*The histogram represents the distribution of snow depth measured with the MagnaProbe. The vertical lines represent the measured snow depth of the different SMP profiles.*

150 Figure 13: The legend can be provided outside the figure and scaled up for clarity.

Legend has been put on the side and the figure has been scaled to fit the full width of the page to improve clarity.

**References**

[revised manuscript text omitted]